# FLOW DISTILLATION SAMPLING: REGULARIZING 3D GAUSSIANS WITH PRE-TRAINED MATCHING PRIORS

**Lin-Zhuo Chen** [*]
Nanjing University

**Kangjie Liu** [*]
Nanjing University

**Youtian Lin**
Nanjing University

**Zhihao Li**
Huawei Noah's Ark Lab

**Siyu Zhu**
Fudan University

**Xun Cao**
Nanjing University

**Yao Yao** [†]
Nanjing University

## ABSTRACT

3D Gaussian Splatting (3DGS) has achieved excellent rendering quality with fast training and rendering speed. However, its optimization process lacks explicit geometric constraints, leading to suboptimal geometric reconstruction in regions with sparse or no observational input views. In this work, we try to mitigate the issue by incorporating a pre-trained matching prior to the 3DGS optimization process. We introduce Flow Distillation Sampling (FDS), a technique that leverages pre-trained geometric knowledge to bolster the accuracy of the Gaussian radiance field. Our method employs a strategic sampling technique to target unobserved views adjacent to the input views, utilizing the optical flow calculated from the matching model (Prior Flow) to guide the flow analytically calculated from the 3DGS geometry (Radiance Flow). Comprehensive experiments in depth rendering, mesh reconstruction, and novel view synthesis showcase the significant advantages of FDS over state-of-the-art methods. Additionally, our interpretive experiments and analysis aim to shed light on the effects of FDS on geometric accuracy and rendering quality, potentially providing readers with insights into its performance. Project page: `https://nju-3dv.github.io/projects/fds`.

## 1 INTRODUCTION

3D Gaussian Splatting (3DGS) (Kerbl et al., 2023) has been widely applied to the field of 3D reconstruction and rendering, including novel view synthesis of static scenes (Kerbl et al., 2023; Yu et al., 2024a), mesh surface reconstruction (Guédon and Lepetit, 2024; Yu et al., 2024b), inverse rendering (Liang et al., 2024; Gao et al., 2023), object detection (Cao et al., 2024), and dynamic 3D reconstruction (Wu et al., 2024; Lin et al., 2024), However, in scenarios with less-observed areas, such as indoor scenes and unbounded scenes, radiance field optimization often suffers from overfitting to these limited input views (Li et al., 2024), resulting in unreliable and corrupted geometry reconstruction.

To mitigate the issue, recent research efforts (Li et al., 2024; Paliwal et al., 2024; Turkulainen et al., 2025) have focused on incorporating geometric priors from input views into the training process, thereby regulating the optimization of radiance field represented by 3D Gaussian points. For instance, DN-Splatter (Turkulainen et al., 2025) integrates sensor depth and normal cues into the reconstruction process. However, sensor depth acquisition is costly, and the depth prior information from pre-trained monocular deep models inevitably suffer from the scale ambiguity (Liu et al., 2023b). While the normal prior provides even better geometric details, the scale ambiguity still exists due to its monocular nature.

In contrast to monocular priors, pairwise matching priors can provide absolute scale information of the scene. In this paper, we introduce **F**low **D**istillation **S**ampling (FDS), an online method for distilling matching prior from a pre-trained optical flow model into the 3DGS training process. FDS aims to enhance the geometry quality of Gaussian radiance field by leveraging the matching prior

---

[*]Equal contribution.
[†]Corresponding author.

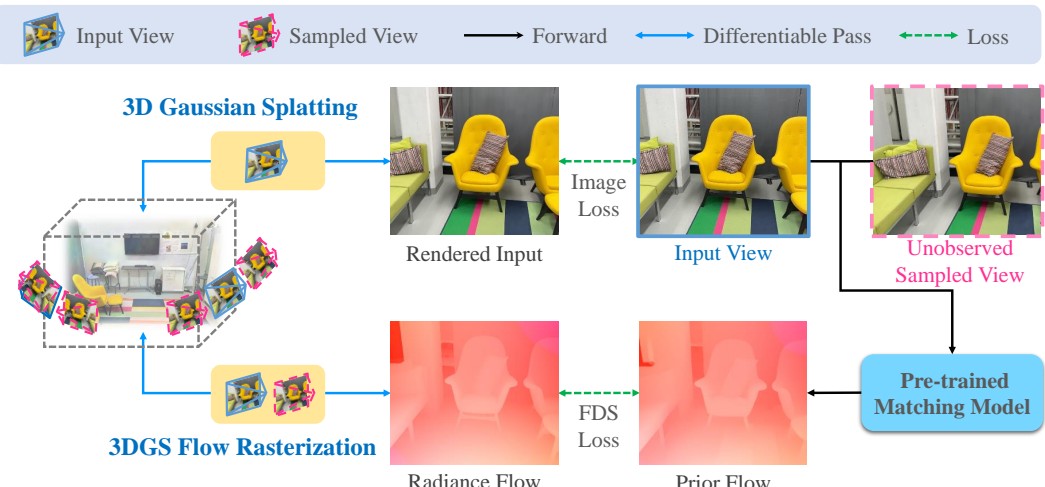

Figure 1. **Pipeline of the proposed FDS.** For each input view, we apply the FDS camera sampling scheme to generate corresponding unobserved sampled view. We then compute Radiance flow base on rendered depth and the Prior flow from matching prior model. Finally the Prior Flow is used to supervise Radiance flow, which enhances the geometric quality of Gaussian Radiance Field.

into the unobserved novel view. Specifically, we observe that the flow between the input view and the unobserved view generated by the match prior model (i.e., **Prior Flow**), can guide and refine the flow analytically calculated from the 3DGS geometry (i.e., **Radiance Flow**), improving the 3DGS reconstruction quality. Moreover, better 3DGS scene will lead to more accurate Prior Flow, creating a mutually reinforcing effect between two computed flow maps. This remains effective even when the radiance field is poorly optimized and the image rendered from the unobserved viewpoint is blurry during training. In addition, a camera sampling scheme is proposed to adaptively control the overlap between input view and sampled view for better Prior Flow calculation, which allows to leverage prior geometric knowledge more profoundly and thereby better enhance the 3DGS reconstruction quality.

The proposed FDS has been extensively evaluated on MushRoom (Ren et al., 2024), ScanNet (V2) (Dai et al., 2017), and Replica (Straub et al., 2019) datasets for the task of geometry reconstruction. We apply FDS to two commonly used baseline approaches, namely 3DGS (Kerbl et al., 2023) and 2DGS (Huang et al., 2024). The results demonstrate a significant improvement in geometry reconstruction accuracy. Additionally, we provide interpretive experiments and comprehensive analysis on the effectiveness of FDS, shedding light on its influence on the quality of geometric reconstruction and novel-view rendering. Our contributions are summarized as follows:

- FDS leverages matching prior information to recover absolute scale, significantly enhancing the geometric quality of the Gaussian radiance field.
- An adaptive camera sampling scheme is proposed to selectively choose unobserved views with controllable overlap with the input view, further improving the geometric quality of the Gaussian radiance field even in less-observed areas.
- By employing FDS and the adaptive camera sampling scheme, we bring significant improvements to state-of-the-art 3D geometry reconstruction approaches.

## 2 RELATED WORK

**Geometry Reconstruction Based on 3DGS**: To enhance the geometry of 3DGS representations, some studies (Gao et al., 2024a; Waczyńska et al., 2024; Lyu et al., 2024; Chen et al., 2023) focus on integrating mesh or SDF representations with 3DGS. (Gao et al., 2024a) improve the splitting of 3DGS under the guidance of mesh representation and deformation gradients. A hybrid representation that integrates 3DGS with mesh, allowing 3DGS to be modified as a mesh, is introduced in (Waczyńska et al., 2024). Mani-GS (Gao et al., 2024b) binds Gaussian and shape-aware triangular

mesh to manipulate 3DGS directly. 3DGSR (Lyu et al., 2024) combines a signed distance field (SDF) network with 3DGS to enhance geometry quality, aligning the SDF network's geometry with that of 3DGS. NeuSG (Chen et al., 2023) utilizes point clouds from 3DGS to regulate NeuS, while its normals are also used to refine 3DGS. Additionally, NeuSG includes regularizers to ensure that 3DGS remains close to the surface. Some work (Guédon and Lepetit, 2024; Huang et al., 2024; Yu et al., 2024b; Chen et al., 2024) tends to extract geometry from 3DGS directly, benefiting from its fast training and rendering speed. Sugar (Guédon and Lepetit, 2024) proposes an efficient mesh extraction method from 3DGS, aligned with a regularization term during the training process and a refinement strategy. 2DGS (Huang et al., 2024) proposed a 2D surface modeling and two regularization losses which can preserve perspective-correct splatting and enchance geometry reconstruction. A ray-tracing-based volume rendering is introduced in GOF (Yu et al., 2024b), allowing the extraction of geometry from 3DGS directly. PGSR (Chen et al., 2024) proposes unbiased depth rendering, combined with singe and multi-view regularization loss to preserve geometric consistency. However, in low-texture and less frequently observed areas, such as indoor scenes, 3DGS still tends to overfit to the limited input views, necessitating regulation through additional prior cues.

**Prior Regulation for Rendering**: Prior information is usually incoporated in ill-posed problems, including sparse view novel view synthesis, dynamic scene reconstruction, and mesh reconstruction. For sparse novel view synthesis tasks, most works utilize the prior depth information (Deng et al., 2022; Roessle et al., 2022; Song et al., 2024; Wang et al., 2023b; Zhu et al., 2023; Xiong et al., 2023; Paliwal et al., 2024), semantic information (Jain et al., 2021; Wynn and Turmukhambetov, 2023; Xiong et al., 2023), and matching information (Paliwal et al., 2024; Lao et al., 2024) to constrain the optimization process of 3DGS. Dynamic scene reconstruction is another challenging task which requires reconstructing the scene geometry and object motion at the same time. Therefore, optical flow priors are crucial as they can help distinguish between camera motion and object motion (Liu et al., 2023b) while providing motion priors between frames (Liu et al., 2023a; Gao et al., 2021; Li et al., 2023; Wang et al., 2023a; Guo et al., 2023; Tian et al., 2023). For the mesh reconstruction task, geometry quality is also enhanced by depth priors (Wei et al., 2021; Yu et al., 2022; Turkulainen et al., 2025) or normal priors (Yu et al., 2022; Turkulainen et al., 2025). However, using optical flow model priors to obtain metric depth priors to help geometric reconstruction, while leveraging unobserved regions to enhance the quality of limited view reconstruction, has not yet been explored.

## 3 METHOD

Our FDS regulates the optimization of Gaussian radiance field by incorporating matching priors from the pretrained deep model. The generation of Radiance Flow and our proposed FDS loss, along with the equipped camera sampling scheme, are detailed in Sec. 3.1 and Sec. 3.2, respectively.

### 3.1 3D GAUSSIAN SPLATTING AND RADIANCE FLOW

We utilize 3DGS as an example to demonstrate how Radiance Flow is generated. 3DGS (Kerbl et al., 2023) employs a collection of 3D Gaussians to represent the 3D scene. The expression of $i$-th 3D Gaussian distribution shows below:

$$\mathcal{G}_i(\boldsymbol{X}) = e^{-\frac{1}{2}(\boldsymbol{X}-\boldsymbol{\mu}_i)^T \boldsymbol{\Sigma}_i^{-1}(\boldsymbol{X}-\boldsymbol{\mu}_i)}, \tag{1}$$

where $\boldsymbol{\Sigma}_i = \boldsymbol{R}_i \boldsymbol{S}_i \boldsymbol{S}_i^T \boldsymbol{R}_i^T$, and $\boldsymbol{\mu}_i$ represents the position of the Gaussian, which is optimized during training. Both $\boldsymbol{S}_i$ and $\boldsymbol{R}_i$ are represented by a 3D scaling vector $\boldsymbol{s}_i$ and a quaternion $\boldsymbol{q}_i$, respectively. In addition to the above parameters, each Gaussian $\boldsymbol{P}_i$ has extra learnable attributes, including opacity $\boldsymbol{\alpha}_i$ and color feature $\boldsymbol{f}_i$.

To render a color for pixel $x$, a volume rendering based method similar to NeRF (Mildenhall et al., 2021) is adapted:

$$\boldsymbol{C}(\boldsymbol{x}) = \sum_{i \in N} \boldsymbol{c}_i \hat{\boldsymbol{\alpha}_i} \prod_{j=1}^{i-1} (1 - \hat{\boldsymbol{\alpha}_j}), \tag{2}$$

where $c_i$ is the color of each Gaussian point, represented by spherical harmonics. $\hat{\alpha}$ is the blending weight for each 2D Gaussian point which is the projection of 3D Gaussian points on the image plane. $N$ is the number of Gaussian points. Similarly, the depth of pixel $x$ is rendered using alpha blending.

$$D(x) = \frac{\sum_{i \in N} d_i \hat{\alpha}_i \prod_{j=1}^{i-1}(1 - \hat{\alpha}_j)}{\sum_{i \in N} \hat{\alpha}_i \prod_{j=1}^{i-1}(1 - \hat{\alpha}_j)}, \tag{3}$$

where $d_i$ is the distance from the target camera to the Gaussian Point $P_i$. We can also render Radiance Flow between two views of camera $m, n$ using their camera poses and position of Gaussian points $\mu_i$. As mentioned above, we can project pixel $x = (u_1, v_1)$ in $m$-th view image to the $n$-th view by its corresponding depth and their pose transformation:

$$D^n(u_2, v_2) \begin{bmatrix} u_2 \\ v_2 \\ 1 \end{bmatrix} = KT_m^n K^{-1} D^m(u_1, v_1) \begin{bmatrix} u_1 \\ v_1 \\ 1 \end{bmatrix}, \tag{4}$$

where $T_m^n$ is the relative transformation from $m$-th view to $n$-th view, $K$ is the intrinsic matrix, and $D^m(u_1, v_1)$ is the rendered depth of $(u_1, v_1)$ in $m$-th view.

Next, we calculate $F^{m \to n}(x)$ for pixel $x = (u_1, v_1)$ from view $m$ to view $n$:

$$F^{m \to n}(x) = \begin{bmatrix} u_2 - u_1 \\ v_2 - v_1 \end{bmatrix}. \tag{5}$$

For other type of Gaussian radiance field such as 2DGS (Huang et al., 2024), we only need to replace the alpha blending based depth $D(x)$ with corresponding formulation.

## 3.2 FLOW DISTILLATION SAMPLING

Given a collection of images $\{I^i\}_{i=1,2,...N}$, Gaussian Radiance Field typically employs the following loss function for rendering optimization:

$$L = \frac{1}{B} \sum_{i=1}^{B} (1 - \lambda)L_1 + \lambda L_{D-SSIM} + \lambda_{normal} L_n, \tag{6}$$

where $B$ denotes batch size, and $L_n$ represents the normal consistency loss from (Huang et al., 2024). However, when $N$ is small, this representation suffers from overfitting (Li et al., 2024; Paliwal et al., 2024).

We propose the Flow Distillation Sampling (FDS) method to incorporate pretrained matching priors into the rendering optimization process, thereby mitigating overfitting and improving rendering performance.

The camera sampling scheme and loss function design in FDS are introduced in Sec. 3.2.1 and Sec. 3.2.2.

### 3.2.1 CAMERA SAMPLING SCHEME

During the training process, FDS randomly samples unobserved camera views nearby the input view and then incorporates the matching priors into these views. To utilize the matching prior information between input view and sampled view

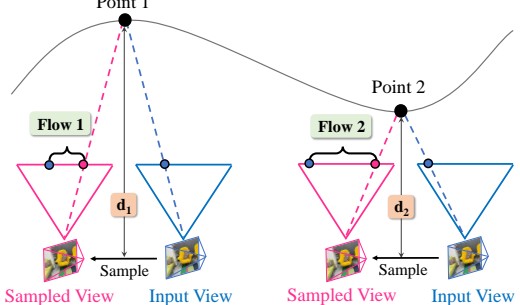

Figure 2. **Explanation of depth-adaptive translation radius.** A fixed-radius camera sampling scheme may result in significantly different flow values (Flow 1 and Flow 2) in areas with varying depth ($d_1$ and $d_2$).

more profoundly, we propose a camera sampling scheme that ensures sufficient movement to perceive the geometry of the scene while avoiding excessively abrupt motions that could make it difficult for the prior model to match.

Given the world to camera transformation matrix $\boldsymbol{T^i}$ for the $i$-th input view, a small translation disturbance $\boldsymbol{t}$ and rotation disturbance $\boldsymbol{R}$ are applied to get the sampled transformation matrix $T^s$, expressed as:

$$\boldsymbol{T^s} = ((\boldsymbol{T^i})^{-1}\boldsymbol{E})^{-1} \tag{7}$$

where $\boldsymbol{E} = [\boldsymbol{R}, \boldsymbol{t}]$ denotes the transformation matrix from the sampled view to the input view, and $t$ is a translation vector satisfying $|\boldsymbol{t}| = \epsilon_t$, $\boldsymbol{t} = (t_1, t_2, t_3)$, indicating that unobserved view is sampled along a small spiral circle with a radius of $\epsilon_t$. As noted in (Bian et al., 2021), the rotation flow in image warping is independent of depth. Therefore, we set the rotation part to identity matrix. So that $E$ is a pure-translation transformation. For the translation radius $\epsilon_t$, as illustrated in Fig. 2, using a uniform radius for all views leads to varying flows due to the differences in depths; the closer the depth, the larger the flow. To ensure flow consistency across all views while maintaining controllable overlap between the input views and sampled views, we hope to implement a depth-adaptive radius, which can help to preserve the same flow between all input views and their sampled views.

According to (4), the pure translation transformation (Bian et al., 2021) between the input view $i$ and its sampled view $s$ is shown below:

$$\boldsymbol{D^s}(u_2, v_2)\begin{bmatrix} u_2 \\ v_2 \\ 1 \end{bmatrix} = \boldsymbol{D^i}(u_1, v_1)\begin{bmatrix} u_1 \\ v_1 \\ 1 \end{bmatrix} + \boldsymbol{K}\begin{bmatrix} t_1 \\ t_2 \\ t_3 \end{bmatrix} \tag{8}$$

where $K$ is the intrinsic matrix of the camera, after solving the above equation (Bian et al., 2021), we get:

$$\begin{bmatrix} u_2 \\ v_2 \end{bmatrix} = \begin{bmatrix} \frac{\boldsymbol{D_i}(u_1,v_1)u_1 + f_x t_1 + c_x t_3}{\boldsymbol{D_i}(u_1,v_1) + t_3} \\ \frac{\boldsymbol{D_i}(u_1,v_1)u_1 + f_y t_2 + c_x t_3}{\boldsymbol{D_i}(u_1,v_1) + t_3} \end{bmatrix} \tag{9}$$

We set $t_3 = 0$ in our camera sampling scheme and assume camera intrinsic parameters: $f_x \approx f_y = f$. The radiance flow $\boldsymbol{F^{i \to s}}(u_1, v_1) = \begin{bmatrix} u_2 - u_1 \\ v_2 - v_1 \end{bmatrix}$ from the input view $i$ to its sampled view $s$ is shown below:

$$\boldsymbol{F^{i \to s}}(u_1, v_1) = \begin{bmatrix} \frac{f}{\boldsymbol{D_i}(u_1,v_1)}t_1 \\ \frac{f}{\boldsymbol{D_i}(u_1,v_1)}t_2 \end{bmatrix} \tag{10}$$

We aim to keep the value of $||\boldsymbol{F^{i \to s}}(u_1, v_1)||_2$ constant for the pixel $x = (u_1, v_1)$ during each camera sampling. By setting $||\boldsymbol{F^{i \to s}}(u_1, v_1)||_2 = \sigma$ and incorporating this with (10), we get:

$$\epsilon_t = \sqrt{t_1^2 + t_2^2} = \sigma \frac{\boldsymbol{D_i}(u_1, v_1)}{f} \tag{11}$$

Thus, the radius of translation in our camera sampling is defined as $\epsilon_t = \sigma \frac{\boldsymbol{D_i}(u_1,v_1)}{f}$ which helps maintain stable flow. The parameter $\sigma$ can be tuned as a hyperparameter, which represents the mean radiance flow between the input view and its sampled view. Given that pixel depths vary within an image, we use the mean depth $\bar{\boldsymbol{D_i}}$ of the image and set the radius of our translation $\epsilon_t = \sigma \frac{\bar{\boldsymbol{D_i}}}{f}$.

$$\boldsymbol{t} = \begin{bmatrix} \sigma \frac{\bar{\boldsymbol{D_i}}}{f} sin(2\pi\xi) \\ \sigma \frac{\bar{\boldsymbol{D_i}}}{f} cos(2\pi\xi) \\ 0 \end{bmatrix} \tag{12}$$

Where $\xi \sim U(0, 1)$ is a uniform random value between $[0, 1]$ during training.

### 3.2.2 FLOW DISTILLATION SAMPLING LOSS

Specifically, to optimize the rendered results of the $i$-th view, every iteration, we sample an unobserved view $s$ near the $i$-th input training camera to render the color image $C^s$. The Radiance Flow $F^{i \to s}(p)$, guided by Equ. (5), benefits from the introduced matching priors. We define the Prior Flow $\overline{F}^{i \to s}$ as:

$$\overline{F}^{i \to s} = \mathcal{M}_\theta(I^i, C^s), \tag{13}$$

where $\theta$ represents the parameters of pretrained network $\mathcal{M}$, $I^i$ is the rgb ground truth of view $i$, and $C^s$ is the rendered image of sampled view $s$. Instead of using $C^i$ to generate Prior Flow, we find that using $I^i$ can help us to remove the floaters in $C^s$ as shown in Tab. 4. However, both $F^{i \to s}$ and $\overline{F}^{i \to s}$ face challenges:

- $F^{i \to s}$, derived from Equ. (5), is inaccurate due to the incorrect positioning of Gaussian points.

- $\overline{F}^{i \to s}$, derived from Equ. (13), suffers from the blurred rendering quality of $C^s$.

Despite these inaccuracies, we observe that $\overline{F}^{i \to s}$ is more robust and precise during training compared with $F^{i \to s}$. Motivated by this observation, we aim to utilize $\overline{F}^{i \to s}$ to refine $F^{i \to s}$, which can make $F^{i \to s}$ more accurate, enhancing the positioning of Gaussian points and subsequently improving the rendering quality of $C^s$. This process also leads to a more accurate $\overline{F}^{i \to s}$. Based on these observations, we propose the FDS loss, which distills matching priors from a pretrained deep model through mutual refinement of the two flows:

$$L_{fds} = ||\overline{F}^{i \to s} - F^{i \to s}||_2 \tag{14}$$

---

**Algorithm 1** Flow Distillation Sampling

---

**Input:** A batch of input training image: $\{I_i\}_{i=1}^N$, Transformation Matrix: $\{T_i\}_{i=1}^N$, Prior Matching Network $\mathcal{M}_\theta$, Gaussian Points $\{P_i\}_{i=1}^M$ with $\{r_i, t_i, f_i, \mu_i, \alpha_i\}$.
**Output:** $L_{fds}$
1: **for** $i$ in $\{1, 2, \dots, B\}$ **do**
2:      $\xi \sim U(0, 1), R \leftarrow I$
3:      $t_1 \leftarrow \sigma \frac{D_i}{f} sin(2\pi\xi), t_2 \leftarrow \sigma \frac{D_i}{f} cos(2\pi\xi), t_3 \leftarrow 0$
4:      $E \leftarrow [R, t], T^s \leftarrow ((T^i)^{-1}E)^{-1}$
5:      $C^s, D^s \leftarrow Render(T^s, P)$
6:      $C^i, D^i \leftarrow Render(T^i, P)$
7:      $X^s \leftarrow KT_i^s K^{-1} D^i(X^i) X^i / D(X^s)$
8:      $F^{i \to s} \leftarrow X^s - X^i$
9:      $\overline{F}^{i \to s} \leftarrow \mathcal{M}_\theta(I^i, C^s)$
10:     $L_{fds} \leftarrow L_{fds} + 1/B||\overline{F}^{i \to s} - F^{i \to s}||_2$
11: **return** $L_{fds}$

---

To maintain training stability and reduce computational complexity, we detach $\overline{F}^{i \to s}$ from $P_i$ when we calculate loss. This prevents the propagation of gradients, as computing them is resource-intensive and time-consuming, as noted in (Poole et al., 2022). Additionally, this helps to ensure that $\overline{F}^{i \to s}$ is not directly influenced by the less reliable $F^{i \to s}$.

In summary, the procedures of our proposed Flow Distillation Sampling are presented in Algorithm 1. The overall training loss is:

$$L = \frac{1}{B} \sum_{i=1}^B (1 - \lambda) L_1 + \lambda L_{D-SSIM} + \lambda_{normal} L_n + \lambda_{fds} L_{fds}. \tag{15}$$

| Input | 3DGS | 2DGS | 3DGS + FDS | 2DGS + FDS | GT (Depth) |
|-------|------|------|------------|------------|------------|

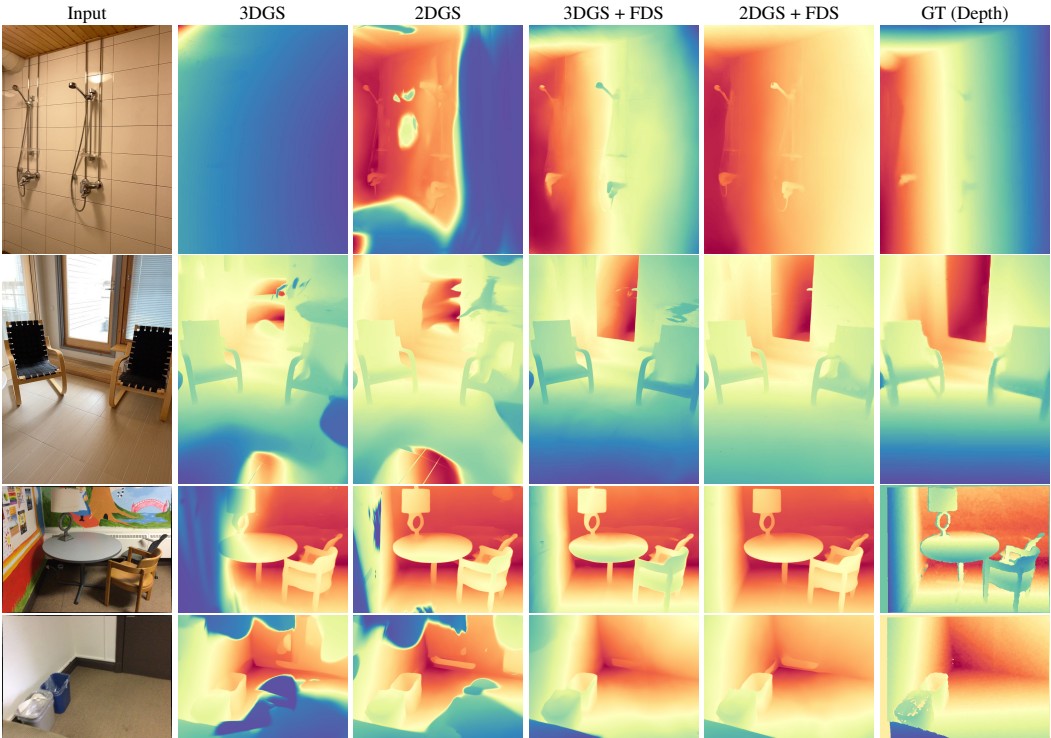

Figure 3. **Comparison of depth reconstruction on Mushroom and ScanNet datasets.** The original 3DGS or 2DGS model equipped with FDS can remove unwanted floaters and reconstruct geometry more preciously.

## 4 EXPERIMENTS

### 4.1 SETUPS

#### 4.1.1 IMPLEMENTATION DETAILS

We apply our FDS method to two types of 3DGS: the original 3DGS, and 2DGS (Huang et al., 2024). The number of iterations in our optimization process is 35,000. We follow the default training configuration and apply our FDS method after 15,000 iterations, then we add normal consistency loss for both 3DGS and 2DGS after 25000 iterations. The weight for FDS, $\lambda_{fds}$, is set to 0.015, the $\sigma$ is set to 23, and the weight for normal consistency is set to 0.15 for all experiments. We removed the depth distortion loss in 2DGS because we found that it degrades its results in indoor scenes. The Gaussian point cloud is initialized using Colmap for all datasets. We tested the impact of using Sea Raft (Wang et al., 2024) and Raft (Teed and Deng, 2020) on FDS performance. Due to the blurriness of the ScanNet dataset, additional prior constraints are required. Thus, we incorporate normal prior supervision on the rendered normals in ScanNet (V2) dataset by default. The normal prior is predicted by the Stable Normal model (Ye et al., 2024) across all types of 3DGS. The entire framework is implemented in PyTorch (Paszke et al., 2019), and all experiments are conducted on a single NVIDIA 4090D GPU.

#### 4.1.2 DATASETS AND METRICS

We evaluate our method for 3D reconstruction and novel view synthesis tasks on **Mushroom** (Ren et al., 2024), **ScanNet (v2)** (Dai et al., 2017), and **Replica** (Straub et al., 2019) datasets, which feature challenging indoor scenes with both sparse and dense image sampling. The Mushroom dataset is an indoor dataset with sparse image sampling and two distinct camera trajectories. We train our model on the training split of the long capture sequence and evaluate novel view synthesis on the test split of the long capture sequences. Five scenes are selected to evaluate our FDS, including "coffee room", "honka", "kokko", "sauna", and "vr room". ScanNet(V2) (Dai et al., 2017) consists of 1,613 indoor scenes with annotated camera poses and depth maps. We select 5 scenes from the ScanNet (V2)

Table 1. **3D Reconstruction and novel view synthesis results on Mushroom dataset. * Represents that FDS uses the Raft model.**

| Method | Acc↓ | Comp↓ | C-L1↓ | NC↑ | F-Score↑ | Abs Rel↓ | PSNR↑ | SSIM↑ | LPIPS↓ | Time↓ |
|---|---|---|---|---|---|---|---|---|---|---|
| GOF | 0.1812 | 0.1093 | 0.1453 | 0.6292 | 0.3665 | 0.2380 | 21.37 | 0.7762 | 0.3132 | ≈1.4h |
| PGSR | 0.0971 | 0.1420 | 0.1196 | 0.7193 | 0.5105 | 0.1723 | 22.13 | 0.7773 | 0.2918 | ≈1.2h |
| 3DGS | 0.1167 | 0.1033 | 0.1100 | 0.7954 | 0.3739 | 0.1214 | 24.18 | 0.8392 | 0.2511 | ≈0.8h |
| 3DGS + FDS* | 0.0569 | 0.0676 | 0.0623 | 0.8105 | 0.6573 | 0.0603 | 24.72 | 0.8489 | 0.2379 | ≈1.3h |
| 3DGS + FDS | **0.0527** | **0.0565** | **0.0546** | **0.8178** | **0.6958** | **0.0568** | **24.76** | **0.8486** | **0.2381** | ≈1.3h |
| 2DGS | 0.1078 | 0.0850 | 0.0964 | 0.7835 | 0.5170 | 0.1002 | 23.56 | 0.8166 | 0.2730 | ≈0.8h |
| 2DGS + FDS* | 0.0689 | 0.0646 | 0.0667 | 0.8042 | 0.6582 | 0.0589 | 23.98 | 0.8255 | 0.2621 | ≈1.3h |
| 2DGS + FDS | **0.0615** | **0.0534** | **0.0574** | **0.8151** | **0.6974** | **0.0561** | **24.06** | **0.8271** | **0.2610** | ≈1.3h |

Table 2. **3D Reconstruction and novel view synthesis results on ScanNet dataset.**

| Method | Acc↓ | Comp↓ | C-L1↓ | NC↑ | F-Score↑ | Abs Rel↓ | PSNR↑ | SSIM↑ | LPIPS↓ |
|---|---|---|---|---|---|---|---|---|---|
| GOF | 1.8671 | 0.0805 | 0.9738 | 0.5622 | 0.2526 | 0.1597 | 21.55 | 0.7575 | 0.3881 |
| PGSR | 0.2928 | 0.5103 | 0.4015 | 0.5567 | 0.1926 | 0.1661 | 21.71 | 0.7699 | 0.3899 |
| 3DGS | 0.4867 | 0.1211 | 0.3039 | 0.7342 | 0.3059 | 0.1227 | 22.19 | 0.7837 | 0.3907 |
| 3DGS + FDS | **0.2458** | **0.0787** | **0.1622** | **0.7831** | **0.4482** | **0.0573** | **22.83** | **0.7911** | **0.3826** |
| 2DGS | 0.2658 | 0.0845 | 0.1752 | 0.7504 | 0.4464 | 0.0831 | 22.59 | 0.7881 | 0.3854 |
| 2DGS + FDS | **0.1457** | **0.0679** | **0.1068** | **0.7883** | **0.5459** | **0.0432** | **22.91** | **0.7928** | **0.3800** |

dataset, uniformly sampling one-tenth of the views, following the approach in (Guo et al., 2022). To further improve the geometry rendering quality of 3DGS, Replica (Straub et al., 2019) contains small-scale real-world indoor scans. We evaluate our FDS on five scenes from Replica: office0, office1, office2, office3 and office4, selecting one-tenth of the views for training. The results for Replica are provided in the supplementary materials. To evaluate the rendering quality and geometry of 3DGS, we report PSNR, SSIM, and LPIPS for rendering quality, along with Absolute Relative Distance (Abs Rel) as a depth quality metrics. Additionally, for mesh evaluation, we use metrics including Accuracy, Completion, Chamfer-L1 distance, Normal Consistency, and F-scores.

## 4.2 Results

### 4.2.1 Depth rendering and novel view synthesis

The comparison results on Mushroom and ScanNet are presented in Tab. 1 and Tab. 2, respectively. Due to the sparsity of sampling in the Mushroom dataset, challenges are posed for both GOF (Yu et al., 2024b) and PGSR (Chen et al., 2024), leading to their relative poor performance on the Mushroom dataset. Our approach achieves the best performance with the FDS method applied during the training process. The FDS significantly enhances the geometric quality of 3DGS on the Mushroom dataset, improving the "abs rel" metric by more than 50%. We found that Sea Raft (Wang et al., 2024) outperforms Raft (Teed and Deng, 2020) on FDS, indicating that a better optical flow model can lead to more significant improvements. Additionally, the render quality of RGB images shows a slight improvement, by 0.58 in 3DGS and 0.50 in 2DGS, benefiting from the incorporation of cross-view consistency in FDS. On the Mushroom dataset, adding the FDS loss increases the training time by half an hour, which maintains the same level as baseline. Similarly, our method shows a notable improvement on the ScanNet dataset as well using Sea Raft (Wang et al., 2024) Model. The "abs rel" metric in 2DGS is improved nearly 50%. This demonstrates the robustness and effectiveness of the FDS method across different datasets.

The qualitative comparisons on the Mushroom and ScanNet dataset are illustrated in Fig. 3. As seen in the first row of Fig. 3, both the original 3DGS and 2DGS suffer from overfitting, leading to corrupted geometry generation. Our FDS effectively mitigates this issue by supervising the matching relationship between the input and sampled views, helping to recover the geometry. FDS also improves the refinement of geometric details, as shown in other rows. By incorporating the matching prior through FDS, the quality of the rendered depth is significantly improved.

### 4.2.2 Mesh extraction

To further demonstrate the improvement in geometry quality, we applied methods used in (Turkulainen et al., 2025) to extract meshes from the input views of optimized 3DGS and 2DGS. The

Table 3. **Ablation study on geometry priors.**

| Method | Acc↓ | Comp↓ | C-L1↓ | NC↑ | F-Score↑ | Abs Rel↓ | PSNR↑ | SSIM↑ | LPIPS↓ |
|---|---|---|---|---|---|---|---|---|---|
| 2DGS | 0.1078 | 0.0850 | 0.0964 | 0.7835 | 0.5170 | 0.1002 | 23.56 | 0.8166 | 0.2730 |
| 2DGS+Depth | 0.0862 | 0.0702 | 0.0782 | 0.8153 | 0.5965 | 0.0672 | 23.92 | 0.8227 | 0.2619 |
| 2DGS+MVDepth | 0.2065 | 0.0917 | 0.1491 | 0.7832 | 0.3178 | 0.0792 | 23.74 | 0.8193 | 0.2692 |
| 2DGS+Normal | 0.0939 | 0.0637 | 0.0788 | **0.8359** | 0.5782 | 0.0768 | 23.78 | 0.8197 | 0.2676 |
| 2DGS+FDS | **0.0615** | **0.0534** | **0.0574** | 0.8151 | **0.6974** | **0.0561** | **24.06** | **0.8271** | **0.2610** |
| 2DGS+Depth+FDS | 0.0561 | 0.0519 | 0.0540 | 0.8295 | 0.7282 | 0.0454 | 24.22 | 0.8291 | 0.2570 |
| 2DGS+Normal+FDS | **0.0529** | **0.0450** | **0.0490** | 0.8477 | **0.7430** | **0.0443** | 24.10 | 0.8283 | 0.2590 |
| 2DGS+Depth+Normal | 0.0695 | 0.0513 | 0.0604 | **0.8540** | 0.6723 | 0.0523 | 24.09 | 0.8264 | 0.2575 |
| 2DGS+Depth+Normal+FDS | **0.0506** | **0.0423** | **0.0464** | **0.8598** | **0.7613** | **0.0403** | **24.22** | **0.8300** | **0.0403** |

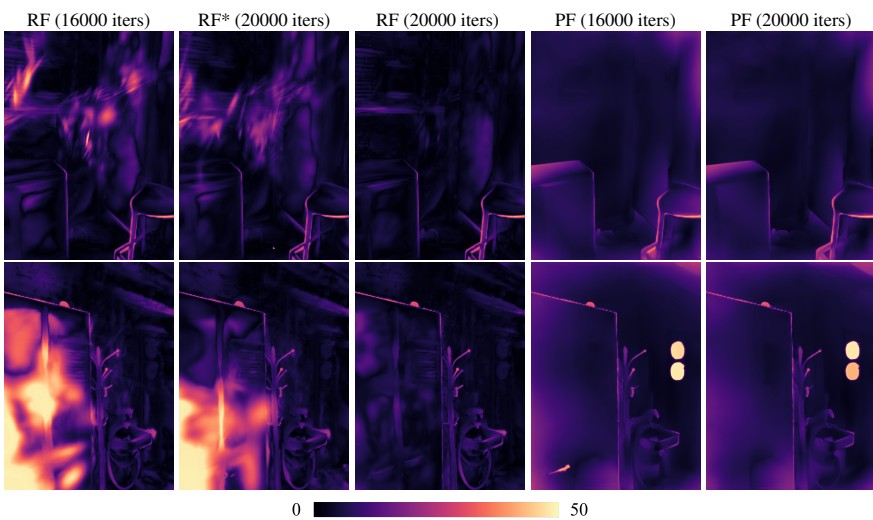

Figure 4. **The error map of Radiance Flow and Prior Flow.** RF: Radiance Flow, PF: Prior Flow, * means that there is no FDS loss supervision during optimization.

comparison results are presented in Tab. 1. With the integration of FDS, the mesh quality is significantly enhanced compared to the baseline, featuring fewer floaters and more well-defined shapes.

## 4.3 ABLATION STUDY

**Ablation study on geometry priors:** To highlight the advantage of incorporating matching priors, we incorporated various types of priors generated by different models into 2DGS. These include a monocular depth estimation model (Depth Anything v2) (Yang et al., 2024), a two-view depth estimation model (Unimatch) (Xu et al., 2023), and a monocular normal estimation model (StableNormal) (Ye et al., 2024). We adapt the scale and shift-invariant loss in Midas (Birkl et al., 2023) for monocular depth supervision and L1 loss for two-view depth supervison. We use Sea Raft (Wang et al., 2024) as our default optical flow model. The comparison results on Mushroom dataset are shown in Tab. 3. We observe that the normal prior provides accurate shape information, enhancing the geometric quality of the radiance field. The multi-view depth prior, hindered by the limited feature overlap between input views, fails to offer reliable geometric information. We test average "Abs Rel" of multi-view depth prior , and the result is 0.19, which performs worse than the "Abs Rel" results rendered by original 2DGS. From the results, it can be seen that depth order information provided by monocular depth improves reconstruction accuracy. Meanwhile, our FDS achieves the best performance among all the priors, and by integrating all three components, we obtained the optimal results.

**Ablation study on FDS:** In this section, we present the design of our FDS method through an ablation study on the Mushroom dataset to validate its effectiveness. The optional configurations of FDS are outlined in Tab. 4. Our base model is the 2DGS equipped with FDS, and its results are shown in the first row. The goal of this analysis is to evaluate the impact of various strategies on FDS sampling and loss design. We observe that when we replace $I_i$ in (13) with $C_i$, as shown in the second row, the geometric quality of 2DGS deteriorates. Using $I_i$ instead of $C_i$ help us to remove the

Table 4. **Ablation study on FDS strategies.**

| $\mathcal{M}_\theta(X, C^s)$ | | Loss | | | Metric | | |
|---|---|---|---|---|---|---|---|
| $X = C^i$ | $X = I^i$ | Next Input view | Sampled view | Fixed Sampled view | Abs Rel ↓ | F-score ↑ | NC ↑ |
| | ✓ | | ✓ | | **0.0561** | **0.6974** | **0.8151** |
| ✓ | | | ✓ | | 0.0839 | 0.6242 | 0.8030 |
| | ✓ | ✓ | | | 0.0877 | 0.6091 | 0.7614 |
| | ✓ | | | ✓ | 0.0724 | 0.6312 | 0.8015 |

floaters in $C^s$, which are also remained in $C^i$. We also experiment with modifying the FDS loss. For example, in the third row, we use the next training input view as the sampling view, and replace the render result of next training view with ground truth image of its input view. Due to the significant movement between images, the Prior Flow fails to accurately match the pixel between them, leading to a further degradation in geometric quality. Finally, we attempt to fix the sampling view and found that this severely damaged the geometric quality, indicating that random sampling is essential for the stability of the mean error in the Prior Flow.

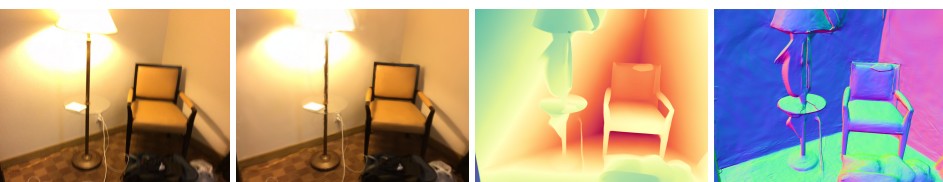

Figure 5. **Limitation of FDS.**

**Interpretive Experiments:** To demonstrate the mutual refinement of two flows in our FDS, for each view, we sample the unobserved views multiple times to compute the mean error of both Radiance Flow and Prior Flow. We use Raft (Teed and Deng, 2020) as our default optical flow model for visualization. The ground truth flow is calculated based on Eq. (4) and Eq. (5) utilizing ground truth depth in dataset. We introduce our FDS loss after 16000 iterations during optimization of 2DGS. The error maps are shown in Fig. 4. Our analysis reveals that Radiance Flow tends to exhibit significant geometric errors, whereas Prior Flow can more accurately estimate the geometry, effectively disregarding errors introduced by floating Gaussian points.

## 4.4 LIMITATION AND FURTHER WORK

Firstly, our FDS faces challenges in scenes with significant lighting variations between different views, as shown in the lamp of first row in Fig. 5. Incorporating exposure compensation into FDS could help address this issue. Additionally, our method struggles with reflective surfaces and motion blur, leading to incorrect matching. In the future, we plan to explore the potential of FDS in monocular video reconstruction tasks, using only a single input image at each time step.

## 5 CONCLUSIONS

In this paper, we propose Flow Distillation Sampling (FDS), which leverages the matching prior between input views and sampled unobserved views from the pretrained optical flow model, to improve the geometry quality of Gaussian radiance field. Our method can be applied to different approaches (3DGS and 2DGS) to enhance the geometric rendering quality of the corresponding neural radiance fields. We apply our method to the 3DGS-based framework, and the geometry is enhanced on the Mushroom, ScanNet, and Replica datasets.

## ACKNOWLEDGEMENTS

This work was supported by National Key R&D Program of China (2023YFB3209702), the National Natural Science Foundation of China (62441204, 62472213), and Gusu Innovation & Entrepreneurship Leading Talents Program (ZXL2024361)

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
