# FLOW DISTILLATION SAMPLING: REGULARIZING 3D GAUSSIANS WITH PRE-TRAINED MATCHING PRIORS

SUPPLEMENTARY MATERIAL

## 1 MORE RESULTS

### 1.1 RESULTS ON REPLICA DATASET

We present the results of applying our FDS on 3DGS and 2DGS model on Replica (Straub et al., 2019) dataset. The results are shown in Table 1. We can see that after introducing FDS, the geometric quality of both 3DGS and 2DGS is significantly improved, especially in "Abs Rel" metric. Since the viewpoints in the Replica Dataset have already provided sufficiently comprehensive observations to the scenes, FDS is unable to further improve the rendering quality of the images. Additionally, we present the visualized comparison of extracted mesh in Fig. 1. Both the quality of geometric reconstruction and the smoothness are significantly improved with the application of our FDS.

Table 1. **3D Reconstruction and Novel view synthesis results on Replica dataset.**

| Method | Acc↓ | Comp ↓ | C-L1 ↓ | NC ↑ | F-Score ↑ | Abs Rel ↓ | PSNR ↑ | SSIM ↑ | LPIPS ↓ |
|---|---|---|---|---|---|---|---|---|---|
| 3DGS | 0.1401 | 0.1667 | 0.1534 | 0.8153 | 0.3139 | 0.0817 | **39.31** | **0.9665** | **0.1491** |
| 3DGS + FDS | **0.0442** | **0.0996** | **0.0719** | **0.8671** | **0.6657** | **0.0192** | 39.00 | 0.9656 | 0.1496 |
| 2DGS | 0.0830 | 0.1334 | 0.1082 | 0.8415 | 0.5690 | 0.0496 | **39.92** | **0.9683** | 0.1396 |
| 2DGS + FDS | **0.0268** | **0.0866** | **0.0567** | **0.8894** | **0.7643** | **0.0112** | 39.81 | 0.9681 | **0.1394** |

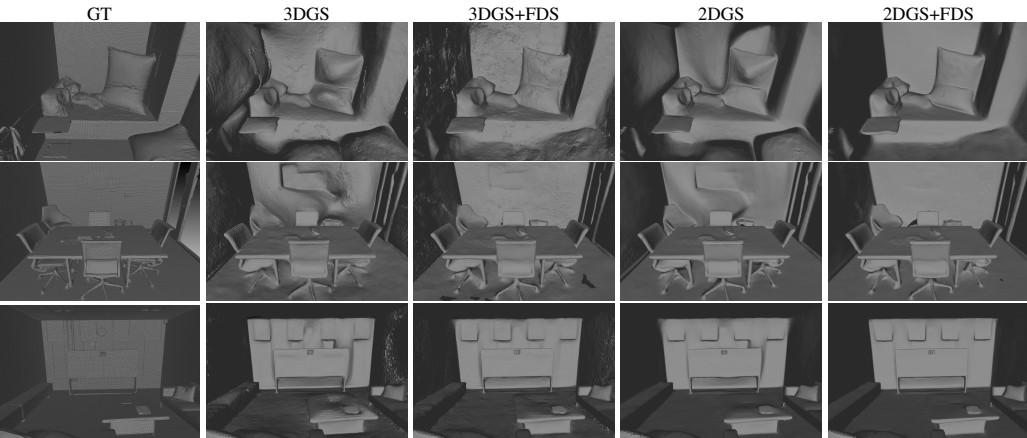

Figure 1. **Qualitative comparison of extracted meshes on Replica datasets.**

### 1.2 RESULTS ON DTU DATASET

DTU (Aanæs et al., 2016) is an object-level dense observation dataset, our FDS is primarily designed to mitigate the issue of insufficient sampling in observation regions. We test results of our FDS on the DTU dataset, as shown in Table 2. While FDS performs better under sparse observations, it still achieves improvements on the DTU dataset with dense observations.

Table 2. **3D Reconstruction on DTU dataset.**

| Method | 24 | 37 | 40 | 55 | 63 | 65 | 69 | 83 | 97 | 105 | 106 | 110 | 114 | 118 | 122 | mean |
|---|---|---|---|---|---|---|---|---|---|---|---|---|---|---|---|---|
| 2DGS | **0.48** | 0.86 | 0.36 | 0.43 | 0.90 | **0.94** | 0.80 | 1.27 | 1.3 | **0.72** | 0.70 | 1.24 | 0.47 | 0.70 | 0.58 | 0.78 |
| 2DGS+FDS | 0.51 | **0.85** | 0.36 | 0.43 | **0.79** | 1.00 | **0.77** | **1.23** | **1.06** | 0.73 | **0.65** | **1.14** | **0.44** | **0.58** | **0.53** | **0.73** |

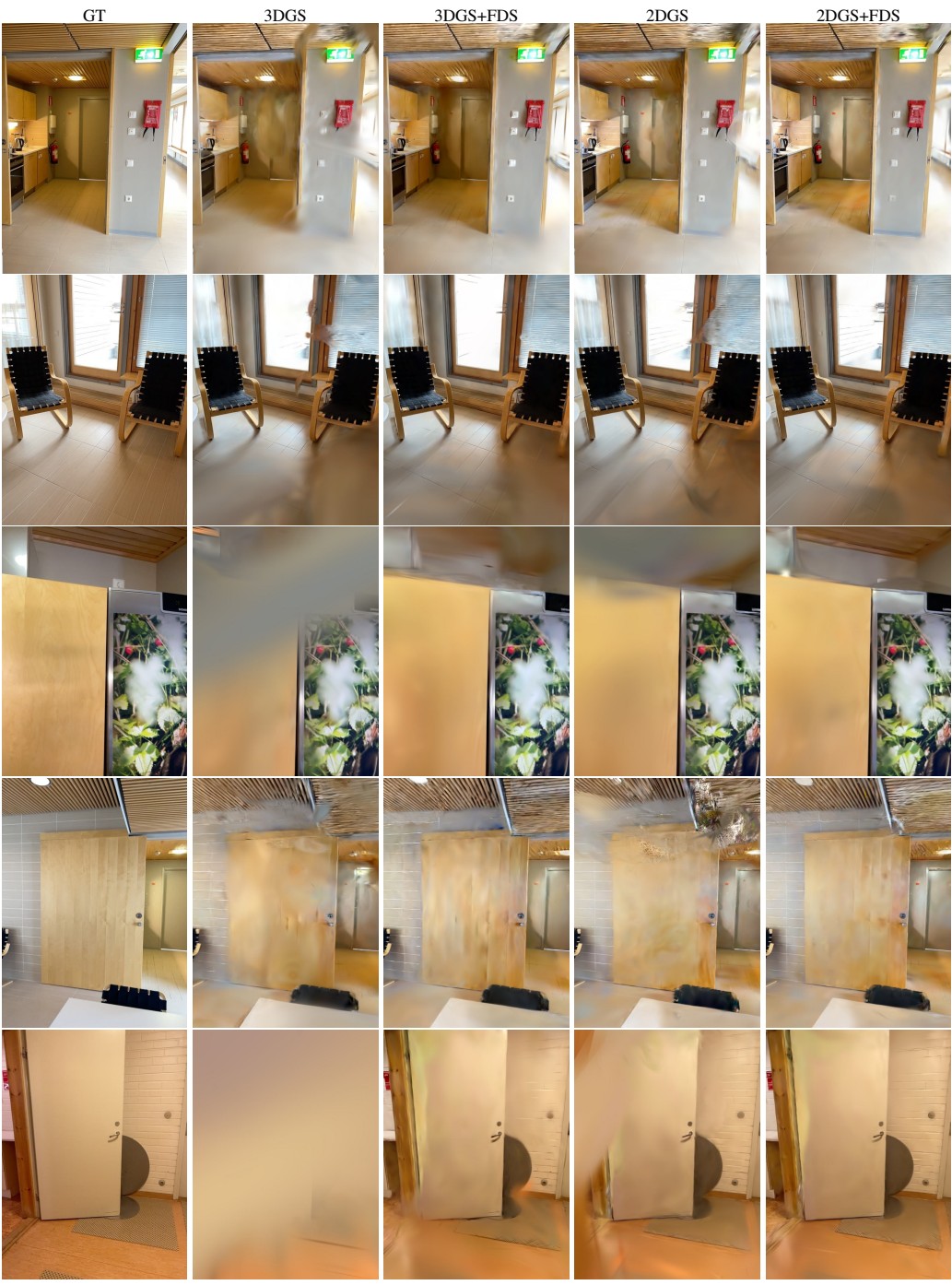

Figure 2. **Qualitative comparison of rendered images on Mushroom (Ren et al., 2024) datasets.**

## 1.3 RENDER QUALITY COMPARISON

We present a visualization of the rendering quality improvements achieved by FDS on the Mushroom dataset, as shown in Fig. 2. It is evident that FDS effectively removes floaters in the air, due to the enhanced consistency between the input and sampling viewpoints introduced by FDS.