# OpenReview forum: "Flow Distillation Sampling: Regularizing 3D Gaussians with Pre-trained Matching Priors"
_ICLR.cc/2025/Conference — ICLR 2025 Poster_

### Official Review · Reviewer_LTym · 2024-10-21

**Soundness:** 3
**Presentation:** 3
**Contribution:** 3
**Rating:** 8
**Confidence:** 4

**Summary:**

In this paper, Flow Distillation Sampling (FDS) is proposed to improve the geometric accuracy and rendering quality of  3D Gaussians Splatting.
FDS first adopts a camera sampling scheme to sample unobserved views near the training views, and then uses the flow predicted by the pre-trained model to guide the flow calculated from the 3DGS geometry.

**Strengths:**

1. The ideas are intuitive, the paper is well written and easy to understand.
2. FDS leverages the matching prior mitigate the overfitting problem and enhance the geometry.

**Weaknesses:**

1. More ablation studies are needed.
- The depth-adaptive radius is calculated by Eq. (8). How to determine the value of the hyperparameter $\sigma$? Is FDS robust to different $\sigma$?
- Is  FDS must require the normal consistency loss $\mathcal{L}_{n}$? In Table 3, how is the performance of “2D-GS+FDS”?
- How to determine the weight for FDS $\lambda_{fds}$?
- How to determine the start iteration (e.g., 15,000) of applying FDS?

2 Lack of visual comparison on the ScanNet dataset.

**Questions:**

See `Weakness`.

---

> ### Author Response · Authors · 2024-11-23
> **Rebuttal by Authors**
>
> **[Q1]:How to determine the value of the hyperparameter $\sigma$?**
>
> **[A1]**:Thank you for your question. we have derived the physical meaning of this hyperparameter $\sigma$,
> which represents the average radius of the 2D flow between the current input view and the unobserved sampling view.
> Therefore, this parameter directly corresponds to a measurable property and can be determined by the flow length that the optical flow model predicts the most accurately, which is typically related to the training data of the optical flow model.
> This parameter can be shared across different types of datasets,
> including DTU datasets (a object centered dataset with dense sampling)
> and Mushroom datasets (an indoor datasets with sparse sampling.)
> We have updated the results of FDS in our paper.
>
> **Derivation:**
>
> As noted in [1], the rotation flow in
> image warping is independent of depth. Therefore, we can set the
> rotation part to identity matrix. so that
> the transformation between the input view $i$ and
> its sampled view $s$ is pure translation, we can change the
> Equ.(4) in our paper to:
>
>
> $$
> D^s(u_2 , v_2) \begin{bmatrix}
>      u_2 \\
>      v_2\\
>      1
>     \end{bmatrix} ^T
> =D^i(u_1 , v_1)
>     \begin{bmatrix}
>      u_1\\
>      v_1\\
>      1
>     \end{bmatrix}^T + K
>     \begin{bmatrix}
>      t_1\\
>      t_2\\
>      t_3
>     \end{bmatrix}^T
> $$
>
> where $K$ is the intrinsic matrix of the camera, after solving the above equation, we get:
>
> $$
> \begin{bmatrix}
>      u_2\\
>      v_2\\
>     \end{bmatrix}^T
> = \begin{bmatrix}
> \frac{D_i(u_1, v_1)u_1+f_xt_1+c_xt_3}{D_i(u_1, v_1) + t_3} \\
> \frac{D_i(u_1, v_1)u_1+f_yt_2+c_xt_3}{D_i(u_1, v_1) + t_3}
> \end{bmatrix}^T
> $$
>
>
> We set $ t_3 = 0 $ in our camera sampling scheme and assume camera intrinsic parameters: $f_x\approx f_y=f$.
> The radiance flow $F^{i\rightarrow s}(u_1, v_1) = \begin{bmatrix}
>      u_2 - u_1\\
>      v_2 - v_1\\
>     \end{bmatrix}^T$from the training view $i$ to its sampled view $s$ is shown below:
>
> $$
> F^{i\rightarrow s}(u_1, v_1) =
>     \begin{bmatrix}
>     \frac{f}{D_i(u_1, v_1)}t_1\\
>     \frac{f}{D_i(u_1, v_1)}t_2
>     \end{bmatrix}
> $$
>
> We aim to keep the value of
> $||F^{i\rightarrow s}(u_1, v_1)||_2$  constant for the pixel $x = (u_1, v_1)$
> during each camera sampling. By setting $||F^{i\rightarrow s}(u_1, v_1)||_2 = \sigma $,
> we get:
>
> $$
>     \epsilon_t = \sqrt{t_1^2 + t_2^2}  =  \sigma  \frac{D_i(u_1, v_1)}{f}
> $$
>
>
>
> Thus, the radius of translation in our camera sampling
> is defined as
> $\epsilon_t = \sigma \frac{D_i(u_1, v_1)}{f} $
> which helps maintain stable flow.
> The parameter $\sigma$ can be tuned as a hyperparameter.
> Given that pixel depths vary within an image,
> we use the mean depth $ \bar{D_i} $ of the image
> and set the radius of our translation
> $\epsilon_t = \sigma \frac{\bar{D_i}}{f} $.
> $||F^{i\rightarrow s}(u_1, v_1)||_2 = \sigma $, demonstrating that $\sigma$
> represents the average radius of the
> 2D flow between the current input view and its unobserved sampling view.
>
> **[Q2]: Is FDS robust to different radius ?**
>
> **[A2]:** As noted in A1, the parameter radius represents
> the average radius of the 2D flow between the current
> input view and its unobserved sampling view.
> So our FDS is robust to different radius in a proper range
> included in the training dataset of optical flow model.
> We added an experiment to validate our statements on the Mushroom dataset.
> The results are shown below:
>
> | Method                        | Acc ↓    | Comp ↓   | C-L1 ↓   | NC ↑     | F-Score ↑ | Abs Rel ↓ | PSNR ↑  | SSIM ↑  | LPIPS ↓  |
> |-------------------------------|----------|----------|----------|----------|-----------|-----------|---------|---------|----------|
> | 2DGS                          | 0.1078   | 0.0850   | 0.0964   | 0.7835   | 0.5170    | 0.1002    | 23.56   | 0.8166  | 0.2730   |
> | 2DGS + FDS (σ=11.5)           | 0.0765   | 0.0593   | 0.0679   | 0.8107   | 0.6469    | 0.0574    | 23.94   | 0.8244  | 0.2633   |
> | 2DGS + FDS (σ=23)             | 0.0615   | **0.0534** | 0.0574   | **0.8151** | 0.6974    | **0.0561** | 24.06   | 0.8271  | 0.2610   |
> | 2DGS + FDS (σ=30)             | **0.0594** | 0.0539   | **0.0566** | 0.8089   | **0.7023** | 0.0571    | **24.09** | **0.8276** | **0.2609** |
>
> It can be observed that both increasing and decreasing the radius allow FDS to achieve a consistent level of improvement.
>
> [1]. Bian, Jia-Wang, et al. "Auto-rectify network for unsupervised indoor depth estimation." IEEE transactions on pattern analysis and machine intelligence 44.12 (2021): 9802-9813.

---

> > ### Author Response · Authors · 2024-11-23
> > **Rebuttal by Authors (2)**
> >
> > **[Q3]: Does FDS require the normal consistency loss ?**
> >
> > **[A3]:** The normal consistency loss is not an essential component of FDS. The normal consistency loss is a regularization loss introduced in 2dgs which can help to align the splats’ normal with the gradients of the depth. Our FDS can also help to improve the reconstruction without this loss. We add the results of 2DGS (w/o nc loss) and FDS+2DGS (w/o nc loss) on the Mushroom dataset:
> >
> > | Method                              | Acc ↓    | Comp ↓   | C-L1 ↓   | NC ↑     | F-Score ↑ | Abs Rel ↓ | PSNR ↑  | SSIM ↑  | LPIPS ↓  |
> > |-------------------------------------|----------|----------|----------|----------|-----------|-----------|---------|---------|----------|
> > | 2DGS                                | 0.1078   | 0.0850   | 0.0964   | 0.7835   | 0.5170    | 0.1002    | 23.56   | 0.8166  | 0.2730   |
> > | 2DGS + FDS                          | **0.0615** | **0.0534** | **0.0574** | **0.8151** | **0.6974** | **0.0561** | **24.06** | **0.8271** | **0.2610** |
> > |-------------------------------------|----------|----------|----------|----------|-----------|-----------|---------|---------|----------|
> > | 2DGS (w/o nc loss)                  | 0.1643   | 0.0904   | 0.1273   | 0.6982   | 0.3853    | 0.1165    | 23.80   | 0.8189  | 0.2627   |
> > | 2DGS (w/o nc loss) + FDS            | **0.0774** | **0.0473** | **0.0624** | **0.7778** | **0.6527** | **0.0578** | **24.32** | **0.8290** | **0.2541** |
> >
> > **[Q4]: In Table 3, how is the performance of “2D-GS+FDS” ?**
> >
> > **[A4]:** The results of "2D-GS+FDS" compared with other prior
> > information on the Mushroom dataset
> > is shown in Table 3 of updated paper.
> >
> > | Method                              | Acc ↓    | Comp ↓   | C-L1 ↓   | NC ↑     | F-Score ↑ | Abs Rel ↓ | PSNR ↑  | SSIM ↑  | LPIPS ↓  |
> > |-------------------------------------|----------|----------|----------|----------|-----------|-----------|---------|---------|----------|
> > | 2DGS                                | 0.1078   | 0.0850   | 0.0964   | 0.7835   | 0.5170    | 0.1002    | 23.56   | 0.8166  | 0.2730   |
> > | 2DGS+Depth                          | 0.0862   | 0.0702   | 0.0782   | 0.8153   | 0.5965    | 0.0672    | 23.92   | 0.8227  | 0.2619   |
> > | 2DGS+Normal                         | 0.0939   | 0.0637   | 0.0788   | 0.8359   | 0.5782    | 0.0768    | 23.78   | 0.8197  | 0.2676   |
> > | 2DGS+FDS                            | **0.0615** | **0.0534** | **0.0574** | **0.8151** | **0.6974** | **0.0561** | **24.06** | **0.8271** | **0.2610** |
> > |-------------------------------------|----------|----------|----------|----------|-----------|-----------|---------|---------|----------|
> > | 2DGS+Depth+FDS                      | 0.0561   | 0.0519   | 0.0540   | 0.8295   | 0.7282    | 0.0454    | **24.22** | **0.8291** | **0.2570** |
> > | 2DGS+Normal+FDS                     | **0.0529** | **0.0450** | **0.0490** | **0.8477** | **0.7430** | **0.0443** | 24.10   | 0.8283  | 0.2590   |
> > | 2DGS+Depth+Normal                   | 0.0695   | 0.0513   | 0.0604   | 0.8540   | 0.6723    | 0.0523    | 24.09   | 0.8264  | 0.2575   |
> > |-------------------------------------|----------|----------|----------|----------|-----------|-----------|---------|---------|----------|
> > | 2DGS+Depth+Normal+FDS               | **0.0506** | **0.0423** | **0.0464** | **0.8598** | **0.7613** | **0.0403** | **24.22** | **0.8300** | **0.0403** |
> >
> > Our “2DGS + FDS” achieves better performance compared to using any other single prior information individually. In combinations with other prior injections, our FDS makes the most significant contribution to performance improvement and can be effectively combined with other prior information to achieve enhanced results.
> >
> > **[Q5]: How to determine the weight for FDS ?**
> >
> > **[A5]:** Similar to other methods supervised by prior information,
> > this weight is determined based on the ratio of values between
> > different loss functions.
> > We set the weight of the FDS loss to be approximately double that of the L1 image loss function.

---

> > > ### Author Response · Authors · 2024-11-23
> > > **Rebuttal by Authors (3)**
> > >
> > > **[Q6]: How to determine the start iteration (e.g., 15,000) of applying FDS ?**
> > >
> > > **[A6]:** We set the start iteration of FDS loss to
> > > 15000 to maintain the same number of points as the baseline for fair comparison.
> > > (The densification of points is stopped after 15000 iterations.)
> > > We tested different starting iterations on the Mushroom dataset and the results are presented below:
> > >
> > > | Method                        | Acc ↓    | Comp ↓   | C-L1 ↓   | NC ↑     | F-Score ↑ | Abs Rel ↓ | PSNR ↑  | SSIM ↑  | LPIPS ↓  |
> > > |-------------------------------|----------|----------|----------|----------|-----------|-----------|---------|---------|----------|
> > > | 2DGS                          | 0.1078   | 0.0850   | 0.0964   | 0.7835   | 0.5170    | 0.1002    | 23.56   | 0.8166  | 0.2730   |
> > > |-------------------------------|----------|----------|----------|----------|-----------|-----------|---------|---------|----------|
> > > | 2DGS + FDS (15000)            | 0.0615   | **0.0534** | **0.0574** | **0.8151** | **0.6974** | 0.0561    | 24.06   | 0.8271  | 0.2610   |
> > > | 2DGS + FDS (10000)            | **0.0590** | 0.0602   | 0.0596   | 0.8053   | 0.6932    | **0.0545** | **24.21** | **0.8300** | **0.2579** |
> > > | 2DGS + FDS (7000)             | 0.0640   | 0.0538   | 0.0589   | 0.8059   | 0.6873    | 0.0584    | 24.13   | 0.8278  | 0.2583   |
> > >
> > >
> > > In our experience, incorporating the FDS loss when PSNR
> > > stabilizes during training yields the best results.
> > > We set the starting iteration of the FDS loss to 15000 to not only maintain the same number of points as the baseline but also stabilize the training PSNR.
> > >
> > > **[Q7]: Lack of visual comparison on the ScanNet dataset.**
> > >
> > > **[A7]:** Thank you for your suggestion. We have added a visual comparison of ScanNet in Figure 3 of the updated paper.

---

> ### Comment · Reviewer_LTym · 2024-11-24
>
> Thank you for your detailed rebuttal. I recommend acceptance of the paper.

---

> > ### Author Response · Authors · 2024-11-25
> >
> > Thank you for your comments and support.

---

### Official Review · Reviewer_w1zd · 2024-10-28

**Soundness:** 3
**Presentation:** 3
**Contribution:** 4
**Rating:** 8
**Confidence:** 4

**Summary:**

This paper aims to improve the 3D Gaussian Splatting reconstruction quality in regions with sparse or no observational input views, by integrating a pre-trained matching prior into the 3DGS optimization process. The matching prior is incorporated through optical flow from a pre-trained model, which supervises the Radiance Flow calculated using 3DGS-rendered depth. Additionally, the authors introduce a Flow Distillation Sampling scheme to efficiently sample unobserved camera views around the input views. The proposed Flow Distillation Loss effectively avoids the scale ambiguity existing in monocular priors. The authors present clear ablation studies and quanlitative improvements to support their claims.

**Strengths:**

- The idea is simple but makes intuitive sense. Matching priors can provide absolute scale information in constrast to monocular priors. This paper offers a promising direction of using pair-wise information prior to advance sparse view reconstruction.
- This ablation study clearly demonstrates the improvement benefited from using this pair-wise matching prior without scale ambiguity. The quanlitative results shown in Table 3. validate the advatange compared to monocular depth prior, even to multi-view depth prior. A clear visualization of the mutual refinement of two flows is also provided.

**Weaknesses:**

- This paper lacks evaluation on widely used geometry reconstruction and novel view synthesis benchmarks such as DTU, Tanks and Temples, and MipNeRF 360. The advantages of the proposed method would be more convincing if the authors could present results on one or more of these benchmarks.
- As mentioned in line 252, both the Prior Flow and Radiance Flow suffer from inaccuracies, raising concerns about the stability of the benefits provided by the proposed Flow Distillation Loss. It is possible that this loss could introduce artifacts or incorrect guidance due to bias. While the metrics in Table 3 appear strong, it’s unclear why the loss significantly outperforms multi-view depth supervision, which does not suffer from inaccurate prior flow. More explanation and analysis are needed to clarify this point.

**Questions:**

- Given that RAFT is computed at every time step, how does the training time for 2DGS + FDS compare to 2DGS?
- Why do both monodepth and multi-view depth seem to only worsen the results, as shown in Table 3.?
- it's said in line 316 that, the normal prior is introduced for evaluation on the ScanNet dataset. But the metrics in Table 2. and Table 3. seem inconsistenct regarding the results of 2DGS + FDS. Does this mean the ScanNet result in Table 2. doesn't use the normal prior?
- Can 2DGS + FDS outperform 2DGS + Normal in Table 3.?

---

> ### Author Response · Authors · 2024-11-23
> **Rebuttal by Authors**
>
> **[Q1]:This paper lacks evaluation on widely used geometry reconstruction and novel view synthesis benchmarks such as DTU, Tanks and Temples, and MipNeRF 360.**
>
> **[A1]:** Thank you for your advice.
> Since DTU is an object-level dense observation dataset,
> our FDS is primarily designed to mitigate the issue of insufficient
> sampling in observation regions.
> Therefore, we did not prioritize testing on the DTU dataset initially.
> We have added the results of our FDS on the DTU dataset, which are shown below.
> While FDS performs better under sparse observations, it still achieves notable improvements on the DTU dataset with dense observations.
>
> | Method        | 24    | 37    | 40    | 55    | 63    | 65    | 69    | 83    |
> |---------------|-------|-------|-------|-------|-------|-------|-------|-------|
> | 2DGS          | **0.48** | 0.86  | 0.36  | 0.43  | 0.90  | **0.94** | 0.80  | 1.27  |
> | 2DGS+FDS      | 0.51  | **0.85** | 0.36  | 0.43  | **0.79** | 1.00  | **0.77** | **1.23** |
>
> | Method        | 97    | 105   | 106   | 110   | 114   | 118   | 122   | mean  |
> |---------------|-------|-------|-------|-------|-------|-------|-------|-------|
> | 2DGS          | 1.30  | **0.72** | 0.70  | 1.24  | 0.47  | 0.70  | 0.58  | 0.78  |
> | 2DGS+FDS      | **1.06** | 0.73  | **0.65** | **1.14** | **0.44** | **0.58** | **0.53** | **0.73** |
>
> **[Q2]:While the metrics in Table 3 appear strong,
> it’s unclear why the loss significantly outperforms multi-view depth supervision,
> which does not suffer from inaccurate prior flow.
> More explanation and analysis are needed to clarify this point.**
>
> **[A2]:** Thank you for your question.
> First, the Mushroom dataset has limited overlap between input views, making multi-view depth estimation unreliable.
> In contrast, our prior flow is extracted from a controlled overlap between the input view and a sampled unobserved view, which is more reliable than offline multi-view depth from uncertain input views.
>
> Second, to reduce inaccuracies caused by relatively blurry sampled views, FDS adopts a random sampling strategy.
> For any single input viewpoint, FDS effectively
> generates a sufficient number of sampled viewpoints
> during training. This acts as a form of model ensemble,
> helping to average out errors.
> To validate this claim, we conducted experiments using fixed sampling viewpoints on the Mushroom dataset instead of random sampling. The results showed
> a significant decline in performance, highlighting the
> importance of this approach.
>
> | Method                        | Acc ↓    | Comp ↓   | C-L1 ↓   | NC ↑     | F-Score ↑ | Abs Rel ↓ | PSNR ↑  | SSIM ↑  | LPIPS ↓  |
> |-------------------------------|----------|----------|----------|----------|-----------|-----------|---------|---------|----------|
> | 2DGS                          | 0.1078   | 0.0850   | 0.0964   | 0.7835   | 0.5170    | 0.1002    | 23.56   | 0.8166  | 0.2730   |
> | 2DGS + FDS (fixed sampling)   | 0.0729   | 0.0617   | 0.0673   | 0.8015   | 0.6312    | 0.0724    | 23.97   | 0.8260  | 0.2623   |
> | 2DGS + FDS                    | **0.0615** | **0.0534** | **0.0574** | **0.8151** | **0.6974** | **0.0561** | **24.06** | **0.8271** | **0.2610** |
>
>
> **[Q3]:Given that RAFT is computed at every time step,
> how does the training time for 2DGS + FDS compare to 2DGS?**
>
> **[A3]:** We have included our training time analysis in Table 1 of the revised paper.
> On the Mushroom dataset, adding the FDS loss led to a half-hour increase in training time. Despite this increase, the overall training time remained comparable to other baselines.
>
> | Method                        | Acc ↓    | Comp ↓   | C-L1 ↓   | NC ↑     | F-Score ↑ | Abs Rel ↓ | PSNR ↑  | SSIM ↑  | LPIPS ↓  | Time  |
> |-------------------------------|----------|----------|----------|----------|-----------|-----------|---------|---------|----------|-------|
> | 2DGS                          | 0.1078   | 0.0850   | 0.0964   | 0.7835   | 0.5170    | 0.1002    | 23.56   | 0.8166  | 0.2730   | 0.8h  |
> | 2DGS + FDS                    | **0.0615** | **0.0534** | **0.0574** | **0.8151** | **0.6974** | **0.0561** | **24.06** | **0.8271** | **0.2610** | 1.3h  |

---

> > ### Author Response · Authors · 2024-11-23
> > **Rebuttal by Authors (2)**
> >
> > **[Q4]: Why do both mono-depth and multi-view depth seem to only worsen the results, as shown in Table 3?**
> >
> > **[A4]:** We initially tested the scale and shift-invariant loss used in
> > Robust Nerf [1] for monocular depth supervision.
> > This method has limited ability to predict absolute depth estimation, which in turn only worsen the result of geometry reconstruction.
> > We have updated our results using a more advanced strategy for monocular depth supervision, as employed in Mono-SDF [2]:
> >
> > $$
> >     L_{depth}=\sum_{r \in R} ||(w^iD^i + q^i) - \hat{D}^{i}||^2
> > $$
> >
> > where $w^i$ and $q^i$ are the scale and shift used to align mono-depth with the absolute depth of input view $i$, calculated using the least squares method.
> > $\hat{D}^{i}$ is monocular depth prediction.
> > The updated results on the Mushroom dataset in Table 3 are presented below.
> >
> > | Method                              | Acc ↓    | Comp ↓   | C-L1 ↓   | NC ↑     | F-Score ↑ | Abs Rel ↓ | PSNR ↑  | SSIM ↑  | LPIPS ↓  |
> > |-------------------------------------|----------|----------|----------|----------|-----------|-----------|---------|---------|----------|
> > | 2DGS                                | 0.1078   | 0.0850   | 0.0964   | 0.7835   | 0.5170    | 0.1002    | 23.56   | 0.8166  | 0.2730   |
> > | 2DGS+Depth                          | 0.0862   | 0.0702   | 0.0782   | 0.8153   | 0.5965    | 0.0672    | 23.92   | 0.8227  | 0.2619   |
> > | 2DGS+Normal                         | 0.0939   | 0.0637   | 0.0788   | 0.8359   | 0.5782    | 0.0768    | 23.78   | 0.8197  | 0.2676   |
> > | 2DGS+FDS                            | **0.0615** | **0.0534** | **0.0574** | **0.8151** | **0.6974** | **0.0561** | **24.06** | **0.8271** | **0.2610** |
> > |-------------------------------------|----------|----------|----------|----------|-----------|-----------|---------|---------|----------|
> > | 2DGS+Depth+FDS                      | 0.0561   | 0.0519   | 0.0540   | 0.8295   | 0.7282    | 0.0454    | **24.22** | **0.8291** | **0.2570** |
> > | 2DGS+Normal+FDS                     | **0.0529** | **0.0450** | **0.0490** | **0.8477** | **0.7430** | **0.0443** | 24.10   | 0.8283  | 0.2590   |
> > | 2DGS+Depth+Normal                   | 0.0695   | 0.0513   | 0.0604   | 0.8540   | 0.6723    | 0.0523    | 24.09   | 0.8264  | 0.2575   |
> > |-------------------------------------|----------|----------|----------|----------|-----------|-----------|---------|---------|----------|
> > | 2DGS+Depth+Normal+FDS               | **0.0506** | **0.0423** | **0.0464** | **0.8598** | **0.7613** | **0.0403** | **24.22** | **0.8300** | **0.0403** |
> >
> > From the results, it can be seen that depth order
> > information provided by monocular depth
> > improves reconstruction accuracy. Meanwhile,
> > our FDS achieves the best performance, and by integrating
> > all three components, we obtained the optimal results.
> > For multi-view depth, we use L1 loss to supervise rendered depth.
> > The results of multi-view depth remain unreliable due to the
> > limited overlap between input views, which is not accounted
> > for in their training datasets. We test average
> > "Abs Rel" of multi-view depth prior. The result is 0.19, which performs
> > worse than the results
> > rendered by original 2DGS whose "Abs Rel" is 0.10.
> >
> > **[Q5]:  Inconsistency regarding the results of 2DGS + FDS between Table 2 and Table 3.**
> >
> > **[A5]:** Thank you for your question.
> > In Table 3, we use the Mushroom dataset for the ablation study,
> > while Table 2 presents the results of our model on the ScanNet dataset which
> > is a different dataset.
> > Therefore, the data in Table 2 and Table 3 are not directly consistent.
> >
> > **[Q6]:  Can 2DGS + FDS outperform 2DGS + Normal in Table 3?**
> >
> > **[A6]:** Thank you for your question. Our 2DGS + FDS can achieve better performance than
> > 2DGS + Normal. The results on the Mushroom dataset are shown below.
> >  We have supplemented more comprehensive results in Table 3. It is worth noting
> > that prior normals can provide more accurate geometric orientations as indicated by 'NC',
> > while our FDS enables more precise geometric positioning as indicated by other metrics.
> > These two improvements are largely complementary to each other.
> >
> > | Method        | Acc ↓    | Comp ↓   | C-L1 ↓   | NC ↑     | F-Score ↑ | Abs Rel ↓ | PSNR ↑  | SSIM ↑  | LPIPS ↓  |
> > |---------------|----------|----------|----------|----------|-----------|-----------|---------|---------|----------|
> > | 2DGS+FDS      | **0.0615** | **0.0534** | **0.0574** | 0.8151   | **0.6974** | **0.0561** | **24.06** | **0.8271** | **0.2610** |
> > | 2DGS+Normal   | 0.0939   | 0.0637   | 0.0788   | **0.8359** | 0.5782    | 0.0768    | 23.78   | 0.8197  | 0.2676   |
> >
> > [1]. Liu, Yu-Lun, et al. "Robust dynamic radiance fields." Proceedings of the IEEE/CVF Conference on Computer Vision and Pattern Recognition. 2023.
> >
> > [2]. Yu, Zehao, et al. "Monosdf: Exploring monocular geometric cues for neural implicit surface reconstruction." Advances in neural information processing systems 35 (2022): 25018-25032.

---

> > > ### Comment · Reviewer_w1zd · 2024-11-26
> > > **Response to the authors**
> > >
> > > Thank you for the detailed response and the additional experiments. All my previous concerns have been addressed. However, I agree with Reviewer 148K that the proposed radiance flow supervision appears not to be limited to 3DGS. It would be valuable to explore whether the proposed methods can be applied to improve NeRF baselines, e.g. Mip-NeRF 360, by addressing issues like floaters. Incorporating these discussions would strengthen the techinical contributions of the paper. I will consider raising my score if the proposed method can also enhance NeRF baselines.

---

> > > > ### Author Response · Authors · 2024-11-27
> > > > **Rebuttal by Authors**
> > > >
> > > > Thank you for your response. Applying our method to NeRF might lead to improvements.
> > > > However, our approach requires rendering entire images of neighboring views, whereas NeRF renders on a per-ray optimization. Rendering full images in NeRF would drastically increase memory consumption and time during training, which in turn makes debugging more challenging and difficult to complete in limited time.
> > > >
> > > > 3DGS is better suited for our method with its higher rendering quality and speed. We have followed the original suggestion from 148K and compared our method with NeRF-based methods. We plan to explore applying FDS to NeRF soon. Thank you for your suggestion!

---

> > > > > ### Comment · Reviewer_w1zd · 2024-11-30
> > > > > **Response to the authors**
> > > > >
> > > > > Thank you for your response. The proposed flow supervision method demonstrates potential for various reconstruction tasks with limited observations, as evidenced by the provided experiments. I will recommend accepting this paper.

---

> ### Author Response · Authors · 2024-12-01
>
> We are very delighted to receive your response and suggestion! Thank you for raising the rating and for your support.

---

### Official Review · Reviewer_148K · 2024-11-03

**Soundness:** 3
**Presentation:** 2
**Contribution:** 2
**Rating:** 6
**Confidence:** 4

**Summary:**

The paper proposes a optical flow based regularization for 3D and 2D Gaussian Splatting. It compares the optical flow output between an input view and a sampled unobserved view to a so-called radiance flow determined from the camera motion and the reconstructed scene. The authors claim that a loss on the difference between the radiance flow an the optical flow results in improved geometry and view synthesis quality. Experiments show improved performance of 3DGS and 2DGS with flow distillation on the SceneNet, Mushroom and Replica dataset.

**Strengths:**

- The paper describes an interesting idea on incorporating a flow prior for 3D reconstruction with Gaussian Splatting.
- The related works contains all relevant geometry reconstruction-based 3DGS works and set them in context to the proposed method.
- The approach is simple and easy to understand, as Figure 1 and 2 are well done and intuitive.

**Weaknesses:**

- The experimental evaluation only considered indoor room datasets, MuSHRoom, ScanNet and Replica. Baseline methods usually use more diverse datasets such as DTU [Jensen et al. 2014], Tanks and Tem-
ples [Knapitsch et al . 2017] and Mip-NeRF360 [Barron et al. 2022].
- The authors found that the depth distortion loss in 2DGS degrades the results. However they do not provide evidence or explanations for that and it is unclear how this influenced the quantitative comparison to 2DGS.
- The paper solely focus on 3DGS-based methods in the related work and also in the experimental evaluation. Comparisons to neural field based methods such as Geo-NeUS or NeuralAngelo would significantly strengthen the claim of state-of-the-art performance.
- The overall quality and the presentation should be improved, e.g. the conclusion is unspecific and contains general claims,  inconsistent capitalization of 'Gaussian', no explanation of \hat{\alpha} in equation 2.

**Questions:**

- The Radiance Flow maps pixels from the source view  to the target view. Considering pixels in the target view containing splatted Gaussians that are occlude in the source view, how is the Radiance Flow computed for these regions?
- In line 267, how does detaching the optical flow influence the overall performance?

**Details Of Ethics Concerns:**

no concerns

---

> ### Author Response · Authors · 2024-11-23
> **Rebuttal by Authors**
>
> **[Q1]: Baseline methods usually use more diverse datasets.**
>
> **[A1]:** Thank you for your advice.
> We did not prioritize testing on the DTU dataset initially since
> DTU is an object-level dense observation dataset, and our FDS
> is primarily designed to mitigate the issue of insufficient
> sampling in observation regions.
> To test our FDS on more diverse datasets,
> We have now included results of our FDS on the DTU dataset, as shown below.
> While FDS performs better under sparse observations, it still achieves notable improvements on the DTU dataset with dense observations.
>
> | Method        | 24    | 37    | 40    | 55    | 63    | 65    | 69    | 83    |
> |---------------|-------|-------|-------|-------|-------|-------|-------|-------|
> | 2DGS          | **0.48** | 0.86  | 0.36  | 0.43  | 0.90  | **0.94** | 0.80  | 1.27  |
> | 2DGS+FDS      | 0.51  | **0.85** | 0.36  | 0.43  | **0.79** | 1.00  | **0.77** | **1.23** |
>
> | Method        | 97    | 105   | 106   | 110   | 114   | 118   | 122   | mean  |
> |---------------|-------|-------|-------|-------|-------|-------|-------|-------|
> | 2DGS          | 1.30  | **0.72** | 0.70  | 1.24  | 0.47  | 0.70  | 0.58  | 0.78  |
> | 2DGS+FDS      | **1.06** | 0.73  | **0.65** | **1.14** | **0.44** | **0.58** | **0.53** | **0.73** |
>
> **[Q2]:  Do not provide evidence or explanations for depth distortion loss
> and it is unclear how this influenced the quantitative comparison to 2DGS.**
>
> **[A2]:** We found that the depth distortion loss tends to
> negatively impact performance on indoor scenes.
> To validate this observation, we evaluated the performance
> of “2DGS (with depth distortion loss)” and “2DGS(with depth distortion loss)+
> FDS”. The results on the Mushroom dataset are shown below.
>
> | Method                          | Acc ↓    | Comp ↓   | C-L1 ↓   | NC ↑     | F-Score ↑ | Abs Rel ↓ | PSNR ↑  | SSIM ↑  | LPIPS ↓  |
> |---------------------------------|----------|----------|----------|----------|-----------|-----------|---------|---------|----------|
> | 2DGS                            | 0.1078   | 0.0850   | 0.0964   | 0.7835   | 0.5170    | 0.1002    | 23.56   | 0.8166  | 0.2730   |
> | 2DGS (distortion loss)          | 0.1225   | 0.1227   | 0.1226   | 0.7621   | 0.4707    | 0.1634    | 22.40   | 0.7916  | 0.3031   |
> | 2DGS (distortion loss) + FDS    | 0.0768   | 0.0705   | 0.0736   | 0.7999   | 0.6511    | 0.0975    | 23.27   | 0.8106  | 0.2815   |
> | 2DGS + FDS                      | **0.0615** | **0.0534** | **0.0574** | **0.8151** | **0.6974** | **0.0561** | **24.06** | **0.8271** | **0.2610** |
>
> For the DTU results in A1, the depth distortion loss is used by default in both 2DGS and 2DGS+FDS. Our FDS also improves reconstruction results.
>
>
> **[Q3]:  Comparisons to neural field based methods such as Geo-NeUS or NeuralAngelo would significantly strengthen the claim of state-of-the-art performance.**
>
> **[A3]:** Thank you for your advice. We are testing NeRF-based methods, such as NeuralAngelo, on the Mushroom dataset.
> We are running the relevant program and will update the results in about two days.
>
> **[Q4]: The overall quality and the presentation should be improved.**
>
> **[A4]:** Thank you for your advice. We have carefully revised our updated paper to improve the presentation quality and address identified errors in method and conclusion section.

---

> > ### Author Response · Authors · 2024-11-23
> > **Rebuttal by Authors (2)**
> >
> > **[Q5]: Considering pixels in the target view containing splatted Gaussians that are occluded in the source view,
> > how is the Radiance Flow computed for these regions ?**
> >
> > **[A5]:** From the source view to the target view, we compute radiance flow using the following equation, based on depth warping:
> >
> > $$
> >     D^n(u_2 , v_2)
> >     \begin{bmatrix}
> >      u_2\\
> >      v_2\\
> >      1
> >     \end{bmatrix}  =  KT_{m}^{n}K^{-1}D^m(u_1 , v_1)
> >      \begin{bmatrix}
> >      u_1\\
> >      v_1\\
> >      1
> >     \end{bmatrix}
> > $$
> >
> > Specifically, we back-project the depth map rendered from the input view into 3D space and calculate the 2D positions of these 3D points in the sampled view using the relative camera poses. The radiance flow is then obtained by subtracting the original pixel positions from these 2D positions. This method allows us to compute the radiance flow for all pixels, including those in self-occluded regions.
> > The accuracy of prior flow tends to be inaccurate in occluded regions, calculated via pixel matching. However, thanks to the random sampling introduced by FDS, for any single training viewpoint, we effectively generate a sufficient number of sampled viewpoints during training.
> > This acts as a form of model ensemble, helping to average out errors in occluded regions, as pixels that are self-occluded in one sampled viewpoint can still be observed from others, thereby mitigating these errors.
> >
> >
> > **[Q6]: In line 267, how does detaching the optical flow influence the overall performance?**
> >
> > **[A6]:** Detaching the optical flow can help to reduce training time and improve performance. We have tested the results of FDS on the Mushroom dataset without detaching the optical flow. We found that the rendered results tend to corrupt, and the training on the 'honka' scene runs out of memory on a 4090D GPU.
> > We report the compared results of the last four scenes:
> >
> > | Method                          | Acc ↓    | Comp ↓   | C-L1 ↓   | NC ↑     | F-Score ↑ | Abs Rel ↓ | PSNR ↑  | SSIM ↑  | LPIPS ↓  |
> > |---------------------------------|----------|----------|----------|----------|-----------|-----------|---------|---------|----------|
> > | 2DGS                            | 0.1078   | 0.0850   | 0.0964   | 0.7835   | 0.5170    | 0.1002    | 23.56   | 0.8166  | 0.2730   |
> > | 2DGS+FDS (without detach)       | 0.1622   | 0.1999   | 0.1811   | 0.6979   | 0.2376    | 0.1680    | 17.23   | 0.6459  | 0.5593   |
> > | 2DGS+FDS                        | **0.0567** | **0.0485** | **0.0526** | **0.8202** | **0.7188** | **0.0509** | **24.47** | **0.8377** | **0.2547** |

---

> ### Author Response · Authors · 2024-11-26
> **Supplemental Details for Question 3**
>
> **[Q3]: Comparisons to neural field based methods such as Geo-NeUS or NeuralAngelo would significantly strengthen the claim of state-of-the-art performance.**
>
> **[A3] (Supplement):** We evaluated the geometric reconstruction performance of the NeuralAngelo method on the Mushroom dataset. We used the original method's data preprocessing script, set the scene type to "indoor," and processed the ground truth camera poses from Mushroom along with the initial point cloud from other experiments in this paper to obtain the bounding sphere of interest. The experimental parameters were configured exactly as in the authors' paper for the TnT dataset, with a batch size of 16 and 500k iterations.
>
> Since the NeuralAngelo method was not designed to reconstruct scene geometry from sparse viewpoints, its performance on Mushroom was suboptimal. Specifically, in the “sauna” scene, using the default preprocessing script caused the NeRF model to diverge. Following the recommendations of the NeuralAngelo authors, we tried adjusting the radius of the bounding sphere of interest. While the model no longer diverged, it tended to overfit to the input viewpoints during training.
>
> To ensure a fair comparison, we exclude the 'sauna' scene, where reconstruction failed by NeuralAngelo, and compare the average results of the remaining four scenes, as shown in the table below:
> | Method          | Acc ↓    | Comp ↓   | C-L1 ↓   | NC ↑     | F-Score ↑ | Time ↓  |
> |-----------------|----------|----------|----------|----------|-----------|--------|
> | **NeuralAngelo** | 0.1016   | 0.0946   | 0.0981   | 0.6505   | 0.4480    |    >128h    |
> | **2DGS**         | 0.1076   | 0.0826   | 0.0951   | 0.7788   | 0.5326    |   **0.8h**  |
> | **2DGS+FDS**     | **0.0635**   | **0.0505**   | **0.0570**   | **0.8102**   | **0.7064**    |  1.3h      |

---

> > ### Author Response · Authors · 2024-11-27
> >
> > We hope our response has addressed your questions. As the discussion phase is coming to a close, we are looking forward to your feedback and would like to know if you have any remaining concerns we can address.
> > We are grateful if you find our revisions satisfactory and consider raising your score for our paper.
> >
> > Thank you once again for the time and effort you have dedicated to reviewing our paper.
> >
> > Best regards
> >
> > Flow Distillation Sampling Authors

---

> > > ### Comment · Reviewer_148K · 2024-11-28
> > >
> > > Thank you for the efforts in the rebuttal. You addressed most of my concern with the DTU experiment the comparison to Neuralangelo and the ablation on the optical flow gradients. For the comparison on DTU you can further compare to NeRF-based methods. All in all, I raise my score to a borderline accept (6).

---

> > > > ### Author Response · Authors · 2024-11-28
> > > >
> > > > Thank you for your response and for improving your rating. We’re delighted to receive and follow your suggestions.

---

### Official Review · Reviewer_GPCw · 2024-11-04

**Soundness:** 2
**Presentation:** 2
**Contribution:** 2
**Rating:** 5
**Confidence:** 5

**Summary:**

the geometric under-constrain problem of 3D Gaussian Splatting (3DGS) in sparse view setups. By incorporating pre-trained matching priors into the optimization process of 3DGS, this method significantly improves both the geometric accuracy and rendering quality of 3DGS.

**Strengths:**

The method proposed in this paper integrates matching priors derived from a pretrained optical flow model to guide the optimization of 3DGS. This approach effectively enhances both the reconstruction quality and rendering quality of existing 3DGS-based methods.

The methodology is well-structured, providing clear and detailed explanations of the proposed FDS technique, the adaptive camera sampling scheme, and the associated loss functions.

Comprehensive experimental evaluations are conducted across multiple datasets, demonstrating the method's effectiveness and robustness.

Additionally, the paper includes interpretive experiments to illustrate the mutual refinement process of the flows, thereby enhancing the understanding of the method's capabilities.

**Weaknesses:**

The results of the proposed method are constrained by the initial quality of the prior flow, however, the reliability of the prior flow cannot be assured under certain sparse viewpoint configurations.

As noted in the limitations section, the method's reliance on the performance of a pretrained optical flow model restricts its generalizability.

While the authors have conducted experiments across multiple datasets, it is important to point out that these datasets are primarily limited to indoor scenes. It is recommended that the authors evaluate their method on a more diverse range of datasets to assess its applicability in various scenarios.

The paper lacks a discussion on the computational complexity of the method. It is recommended that the authors include a detailed report on the training and inference times of the model in the experimental section, along with comparative metrics against other existing methods.

**Questions:**

In the related work section, the review of existing prior art aimed at improving 3DGS performance should be more comprehensive and clearer, particularly concerning the relevant work on optical flow priors. Additionally, in the subsection on Prior Regulation for Rendering, there are sentences with grammatical errors that require careful review and correction.

In Algorithm 1, there are notation errors that need careful checking and correction.

The comparison of depth reconstruction experiments needs to be supplemented with results from other methods to validate the superiority of the proposed approach in geometric reconstruction.

In the dataset section, the paper mentions that the authors have evaluated their method on the Replica dataset, but the experimental results are not presented.

The authors provide limited comparisons with existing methods; it is recommended that they include more baseline methods for a more robust evaluation.

---

> ### Author Response · Authors · 2024-11-23
>
> **[Q1]: The reliability of the prior flow cannot be assured under certain sparse viewpoint configurations.**
>
> **[A1]:** Our FDS introduces two strategies to improve the reliability of prior flow.
> The first is controlled overlap between the **input view** and its
> **sampled unobserved view**,
> which ensures controlled overlap and average flow.
> According to the derivation in our updated paper,
> the hyperparameter $\sigma$ in FDS represents the average radius of the
> 2D flow between the current input view and sampled view.
> This ensures that the optical flow model can generate accurate optical flow under suitable overlap.
> Secondly,
> the random sampling introduced by FDS also provides an advantage. FDS effectively generates a
> sufficient number of sampled viewpoints during training which can help to average out errors. We tested our FDS by fixing the sampling viewpoints instead of using random sampling on the Mushroom dataset. The results are shown below:
>
> | Method                        | Acc ↓    | Comp ↓   | C-L1 ↓   | NC ↑     | F-Score ↑ | Abs Rel ↓ | PSNR ↑  | SSIM ↑  | LPIPS ↓  |
> |-------------------------------|----------|----------|----------|----------|-----------|-----------|---------|---------|----------|
> | 2DGS                          | 0.1078   | 0.0850   | 0.0964   | 0.7835   | 0.5170    | 0.1002    | 23.56   | 0.8166  | 0.2730   |
> | 2DGS + FDS (fixed sampling)   | 0.0729   | 0.0617   | 0.0673   | 0.8015   | 0.6312    | 0.0724    | 23.97   | 0.8260  | 0.2623   |
> | 2DGS + FDS                    | **0.0615** | **0.0534** | **0.0574** | **0.8151** | **0.6974** | **0.0561** | **24.06** | **0.8271** | **0.2610** |
>
> We can see that the random sampling further helps 2DGS improve geometric accuracy.
>
> **[Q2]: The method's reliance on the performance of a pretrained optical flow model restricts its generalization.**
>
> **[A2]:**  Although our model is limited by the accuracy of the optical flow model, the upper bound of our FDS will continue to improve with the emergence of more annotated data and larger models.
> To validate our idea, we replae RAFT with the more advanced SEA-RAFT model in our updated paper, the accuracy of geometric reconstruction has been further enhanced.
>
> | Method                        | Acc ↓    | Comp ↓   | C-L1 ↓   | NC ↑     | F-Score ↑ | Abs Rel ↓ | PSNR ↑  | SSIM ↑  | LPIPS ↓  |
> |-------------------------------|----------|----------|----------|----------|-----------|-----------|---------|---------|----------|
> | 2DGS + FDS (Raft)            | 0.0689   | 0.0646   | 0.0667   | 0.8042   | 0.6582    | 0.0589    | 23.98   | 0.8255  | 0.2621   |
> | 2DGS + FDS (Sea Raft)        | **0.0615** | **0.0534** | **0.0574** | **0.8151** | **0.6974** | **0.0561** | **24.06** | **0.8271** | **0.2610** |
>
> **[Q3]: It is important to point out that these datasets are primarily limited to indoor scenes.**
>
> **[A3]:** To test our FDS on more diverse datasets, we have added results of
> our FDS on DTU datasets shows below. Although our FDS is primarily designed to mitigate the issue
> of insufficient sampling in observation region,
> we still achieve improvements on the DTU dataset with dense observation.
>
> | Method        | 24    | 37    | 40    | 55    | 63    | 65    | 69    | 83    |
> |---------------|-------|-------|-------|-------|-------|-------|-------|-------|
> | 2DGS          | **0.48** | 0.86  | 0.36  | 0.43  | 0.90  | **0.94** | 0.80  | 1.27  |
> | 2DGS+FDS      | 0.51  | **0.85** | 0.36  | 0.43  | **0.79** | 1.00  | **0.77** | **1.23** |
>
> | Method        | 97    | 105   | 106   | 110   | 114   | 118   | 122   | mean  |
> |---------------|-------|-------|-------|-------|-------|-------|-------|-------|
> | 2DGS          | 1.30  | **0.72** | 0.70  | 1.24  | 0.47  | 0.70  | 0.58  | 0.78  |
> | 2DGS+FDS      | **1.06** | 0.73  | **0.65** | **1.14** | **0.44** | **0.58** | **0.53** | **0.73** |

---

> ### Author Response · Authors · 2024-11-23
> **Official Comment by Authors (2)**
>
> **[Q4]: It is recommended that the authors include a detailed
> report on the training and inference times of the model in the experimental section,
> along with comparative metrics against other existing methods.**
>
> **[A4]:** Thank you for your advice. We have included our training time analysis in Table 1 of the revised paper.
> On the Mushroom dataset, adding the FDS loss led to a half-hour increase in training time. Despite this increase, the overall training time remained comparable to other baselines.
> Since FDS is only applied during the training process, it does not affect the inference time.
> | Method           | Acc ↓    | Comp ↓   | C-L1 ↓   | NC ↑     | F-Score ↑ | Abs Rel ↓ | PSNR ↑  | SSIM ↑  | LPIPS ↓  | Time  |
> |------------------|----------|----------|----------|----------|-----------|-----------|---------|---------|----------|-------|
> | GOF              | 0.1812   | 0.1093   | 0.1453   | 0.6292   | 0.3665    | 0.2380    | 21.37   | 0.7762  | 0.3132   | 1.4h  |
> | PGSR             | 0.0971   | 0.1420   | 0.1196   | 0.7193   | 0.5105    | 0.1723    | 22.13   | 0.7773  | 0.2918   | 1.2h  |
> |------------------|----------|----------|----------|----------|-----------|-----------|---------|---------|----------|-------|
> | 3DGS             | 0.1167   | 0.1033   | 0.1100   | 0.7954   | 0.3739    | 0.1214    | 24.18   | 0.8392  | 0.2511   | 0.8h  |
> | 3DGS + FDS       | **0.0527** | **0.0565** | **0.0546** | **0.8178** | **0.6958** | **0.0568** | **24.76** | **0.8486** | **0.2381** | 1.3h  |
> |------------------|----------|----------|----------|----------|-----------|-----------|---------|---------|----------|-------|
> | 2DGS             | 0.1078   | 0.0850   | 0.0964   | 0.7835   | 0.5170    | 0.1002    | 23.56   | 0.8166  | 0.2730   | 0.8h  |
> | 2DGS + FDS       | **0.0615** | **0.0534** | **0.0574** | **0.8151** | **0.6974** | **0.0561** | **24.06** | **0.8271** | **0.2610** | 1.3h  |
>
> **[Q5]: The review of existing prior art aimed at improving 3DGS performance should be more comprehensive and clearer.
> There are sentences with grammatical errors that require careful review and correction.
> In Algorithm 1, there are notation errors that need careful checking and correction.**
>
> **[A5]:** Thank you for your advice. We have carefully reviewed and revised our paper in the related work and methods part.
> In our updated paper, we used colors to highlight the changes.
> Specifically, we adopted a more concise and clear discussion approach in the related work section and added a discussion on the application of priors in optical flow models.
> In the method section (including Algorithm 1), we used more rigorous notation and corrected some errors.
>
> **[Q6]: The comparison of depth reconstruction experiments needs to be supplemented with results from other methods,
>  it is recommended that they include more baseline methods for a more robust evaluation.**
>
> **[A6]:** Thank you for your advice.
> We have compared the depth reconstruction results in Table 1 and Table 2 of our paper, where a lower "Abs Rel" metric indicates better reconstruction quality. Compared with GOF, PGSR, and 2DGS, which
> are SOTA 3DGS-based 3D reconstruction methods, our FDS achieves
> the best depth reconstruction performance.
> We choose 2DGS as the baseline method for our FDS because it performed the best on the Mushroom dataset compared with
> GOF and PGSR.
>
> **[Q7]: In the dataset section, the paper mentions that the authors have evaluated their method on the Replica dataset, but the experimental results are not presented.**
>
> **[A7]:** The experimental results of the Replica dataset are shown in our supplementary materials due to page limitations.
> We also updated our Replica results using a more advanced optical flow model.

---

> ### Author Response · Authors · 2024-11-27
>
> We hope our response has addressed your questions. As the discussion phase is coming to a close, we are looking forward to your feedback and would like to know if you have any remaining concerns we can address.
> We are grateful if you find our revisions satisfactory and consider raising your score for our paper.
>
> Thank you once again for the time and effort you have dedicated to reviewing our paper.
>
> Best regards
>
> Flow Distillation Sampling Authors

---

> ### Author Response · Authors · 2024-12-01
> **Official Comment by Authors**
>
> We would like to sincerely thank you for your valuable feedback and the time you've dedicated to reviewing our paper. As the extended discussion phase is now nearing its end, we would greatly appreciate your feedback on our revisions. Please let us know if there are any remaining issues we can address.
>
> Thank you once again for your support and effort!
>
> Best regards
>
> Flow Distillation Sampling Authors

---

### Author Response · Authors · 2024-11-23
**Rebuttal by Authors**

We sincerely thank all the reviewers for their constructive and insightful advice.
We are encouraged that all reviewers consider the proposed FDS intuitive and interesting.
- Reviewer w1zd: The idea is simple but makes intuitive sense. This paper offers a promising direction of using pair-wise information prior to advance sparse view reconstruction.
- Reviewer LTym: The ideas are intuitive, the paper is well written and easy to understand.
- Reviewer 148K: The paper describes an interesting idea on incorporating a flow prior for 3D reconstruction with Gaussian Splatting.
- Reviewer GPCw: This approach effectively enhances both the reconstruction quality and rendering quality of existing 3DGS-based methods.

Meanwhile, we acknowledge the reviewers' primary concern that
this paper lacks evaluation on some widely used geometry reconstruction dataset, such
as DTU.
Since DTU is an object-level dense observation dataset, our FDS is
primarily designed to mitigate the issue of insufficient sampling in observation regions.
Therefore,
we did not prioritize testing on the DTU dataset initially.
We have now included results of our FDS on the DTU dataset in our supplementary material, as shown below.
While FDS performs better under sparse observations, it still achieves notable improvements on the DTU dataset with dense observations.

| Method        | 24    | 37    | 40    | 55    | 63    | 65    | 69    | 83    |
|---------------|-------|-------|-------|-------|-------|-------|-------|-------|
| 2DGS          | **0.48** | 0.86  | 0.36  | 0.43  | 0.90  | **0.94** | 0.80  | 1.27  |
| 2DGS+FDS      | 0.51  | **0.85** | 0.36  | 0.43  | **0.79** | 1.00  | **0.77** | **1.23** |

| Method        | 97    | 105   | 106   | 110   | 114   | 118   | 122   | mean  |
|---------------|-------|-------|-------|-------|-------|-------|-------|-------|
| 2DGS          | 1.30  | **0.72** | 0.70  | 1.24  | 0.47  | 0.70  | 0.58  | 0.78  |
| 2DGS+FDS      | **1.06** | 0.73  | **0.65** | **1.14** | **0.44** | **0.58** | **0.53** | **0.73** |

We have updated the submitted paper based on each reviewer's suggestions. The main updates include:
- We replaced the original RAFT model with a more advanced optical flow model, SEA-RAFT, enabling FDS to achieve a significant improvement in its performance. The experimental results presented in the submitted paper have all been updated accordingly.
- We have updated Table 3 to include more comprehensive ablation experiments on prior supervision.
- We have added detailed derivations to the paper to clarify the physical meaning of the hyperparameter $\sigma$, providing valuable insights for its configuration.

We have carefully addressed the additional concerns
raised by each reviewer in the corresponding official comments.
Finally, we extend our heartfelt gratitude to all reviewers
and remain open to further
suggestions to improve all aspects of our work.

---

### Meta-Review · Area_Chair_r8B3 · 2024-12-20

**Metareview:**

The paper presents a method that integrates pretrained matching priors to guide the optimization of 3D Gaussian Splatting (3DGS). This approach leverages pretrained geometric knowledge to effectively enhance both the reconstruction and rendering quality of 3DGS methods. Incorporating a flow prior to 3D reconstruction with Gaussian Splatting is interesting and intuitive. The primary concern in the original submission was the lack of experiments on popular datasets such as DTU and MipNeRF. The rebuttal addressed this by including additional experiments on the DTU dataset, strengthening the evaluation of the proposed method.

**Additional Comments On Reviewer Discussion:**

Several issues were raised in the initial reviews, and the rebuttal effectively addressed most of them.

A shared concern is about the limitations of the datasets. In particular, there is a lack of evaluation of some widely used geometry reconstruction datasets, such as DTU. The rebuttal addressed this by including additional experiments on the DTU dataset, where the proposed method demonstrated notable improvements. Additionally, at the reviewer's suggestion, the rebuttal provided comparisons with neural field-based methods, further strengthening the evaluation.

Another concern was the method's dependency on the performance of the pretrained optical flow model. The rebuttal argued that advancements in optical flow models would continuously benefit the proposed method. This was demonstrated by replacing RAFT with SEA-RAFT, which improved performance for the proposed method, illustrating its adaptability to better flow models.

The rebuttal effectively addressed most of the concerns raised during the review process. Most reviewers were optimistic about the paper by the end of the discussion stage.

---

### Decision · Program_Chairs · 2025-01-22

Accept (Poster)